# Enhancing stability by trapping palladium inside N-heterocyclic carbene-functionalized hypercrosslinked polymers for heterogeneous C-C bond formations

Chengtao Yue[1,2,4], Qi Xing[3,4], Peng Sun[1], Zelun Zhao[1], Hui Lv[1] & Fuwei Li [1✉]

Catalyst deactivation caused by the aggregation of active metal species in the reaction process poses great challenges for practical applications of supported metal catalysts in solid-liquid catalysis. Herein, we develop a hypercrosslinked polymer integrated with N-heterocyclic carbene (NHC) as bifunctional support to stabilize palladium in heterogeneous C-C bond formations. This polymer supported palladium catalyst exhibits excellent stability in the one-pot fluorocarbonylation of indoles to four kinds of valuable indole-derived carbonyl compounds in cascade or sequential manner, as well as the representative Suzuki-Miyaura coupling reaction. Investigations on stabilizing effect disclose that this catalyst displays a molecular fence effect in which the coordination of NHC sites and confinement of polymer skeleton contribute together to stabilize the active palladium species in the reaction process. This work provides new insight into the development of supported metal catalysts with high stability and will also boost their efficient applications in advanced synthesis.

[1] State Key Laboratory for Oxo Synthesis and Selective Oxidation, Suzhou Research Institute of LICP, Lanzhou Institute of Chemical Physics (LICP), Chinese Academy of Sciences, Lanzhou, China. [2] University of Chinese Academy of Sciences, Beijing, China. [3] BayRay Innovation Center, Shenzhen Bay Laboratory, Shenzhen, China. [4] These authors contributed equally: Chengtao Yue, Qi Xing. ✉email: fuweili@licp.cas.cn

Palladium (Pd)-catalyzed C–C bond formation reactions, such as Suzuki-Miyaura, Heck, Sonogashira and carbonylation reactions, have been ranked among the most useful and versatile tools for modern organic synthesis[1–5]. Compared with the molecular Pd complex-based homogeneous catalysts, supported Pd catalysts could advance these reactions toward more sustainable conditions due to the easy separation and reuse of the catalyst in the solid–liquid catalysis[6,7]. The "Cocktail" type nature, in terms of complicated catalyst evolution processes (such as dissolution, aggregation, and redeposition) and various metal species (molecular complexes, clusters, and nanoparticles), has been found to be involved in these supported Pd-catalyzed C–C bond formation reactions[8,9]. Unlike the common heterogeneous reactions, wherein the reactants chemically absorb on the catalyst surface and undergo the activation/transformation, these reactions experience a Pd dissolution and redeposition process (Fig. 1). Typically, the detachment of Pd from the catalysts by catalytic interaction[10–12], such as the oxidative addition of aryl halides (ArX) to Pd(0) species, initiates the reactions of C–C bond formation in solution[13,14], and the redeposition of the regenerated Pd(0) species from reductive elimination back onto the support after the completion of the reaction recovers the supported Pd[15]. However, the regenerated Pd(0) species are very unstable and tend to aggregate either in reaction solution or in the process of redeposition, yielding a catalyst quite different from the fresh one[16–18], and then resulting in a decline in the catalytic activity and undesirable recyclability, which poses a major issue for the practical application of supported Pd catalysts in solid–liquid catalysis.

To suppress the Pd aggregation, many nanostructured inorganic and organic supports modified with N-, S-, and P-containing groups have been developed to stabilize the active Pd species through enhancing the metal-support interactions[19,20]. However, these approaches mainly focused on the control of the Pd aggregation over the catalyst surface, and the Pd aggregation in solution has not been effectively prohibited, which still inevitably led to a remarkable deactivation and frustrated recyclability although these Pd aggregates could be redeposited onto the catalyst surface[10,11,21–26]. Thus, the development of a novel support, which can not only efficiently recapture the dissolved Pd before their aggregation in solution but also stabilize the resupported Pd species during the redeposition process, is highly desired to improve the stability of the supported Pd catalysts.

The facilely accessible knitting aryl network polymer (KAP) integrated with strong electron-donating coordination sites are expected to meet the above requirements[27], since these hypercrosslinked KAPs have cage-like polymer skeletons which are supposed to work as molecular fences to efficiently isolate the active Pd species and prevent them from migration/aggregation[28,29], while a built-in monomer bearing electron-rich group can effectively recapture and stabilize the dissolved Pd species[30]. Notably, N-heterocyclic carbenes (NHCs) have recently been recognized as outstanding ligands to coordinate with molecular metal complexes or electronically modify surface metal nanoparticles for efficient homogeneous and heterogeneous catalysis[31–33]. Therefore, it can be speculated that the integration of confinement and coordination effect enabled by the NHC-based hypercrosslinked KAPs could lead to the development of a series of new supported Pd catalysts with enhanced stability for heterogeneous C–C bond formations in solid–liquid catalysis.

The carbonylation reactions represent an important class of C–C bond formation reactions which can introduce carbonyl group into molecules and produce a variety of functional compounds[34]. While ArX are widely used in Pd-catalyzed carbonylation reactions, direct and heterogeneous carbonylation of unfunctionalized Ar–H has been rarely investigated due to the inertness of aryl C–H bond and poor catalyst stability under harsh and oxidative conditions[35]. In this work, we develop a series of KAPs-supported Pd complex precatalysts with different crosslinking structures (KAP-Pd-PEPPSI-x, x = 1, 2, 3, Fig. 2a), which show remarkable activities in carbonylation of the in situ formed 3-iodo-indole via iodine-oxidation of indole C3–H bond to corresponding valuable acyl fluoride, amides, aroyl azides, and aroyl cyanides with significantly enhanced stability, surpassing other conventional supported Pd catalysts. Investigations on the stabilizing effects reveal that the KAP-Pd-PEPPSI display a molecular fence effect in which the coordination of the NHC sites and confinement of the hypercrosslinked polymer skeleton work together to trap the active Pd species inside the swollen polymer in the reaction process, which effectively decrease the Pd content in solution and therefore prevent them from aggregation. Likewise, KAP-Pd-PEPPSI also show superior catalytic stability over traditional Pd@C in the representative Suzuki-Miyaura reaction. This work not only provides a facile entry to develop stable heterogeneous catalyst for solid–liquid catalysis, but also establishes an efficient route to synthesize various indole-derived carbonyl compounds, which have wide applications in the preparation of bioactive compounds.

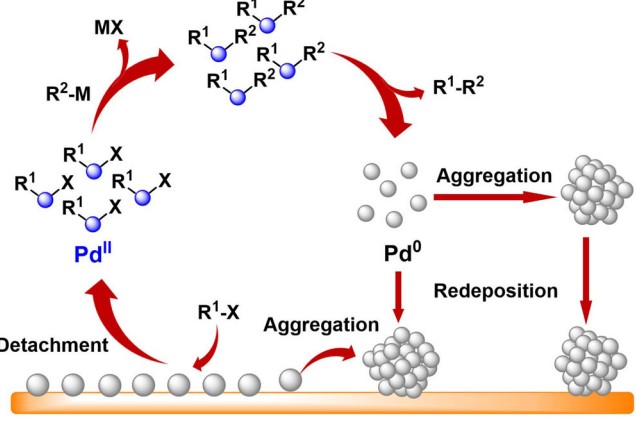

**Fig. 1 Schematic illustration of catalyst evolution.** The "Cocktail" type nature and evolution of the catalytic species in C–C bond formation reactions catalyzed by supported Pd catalysts.

## Results

**Catalyst synthesis and characterization.** First, the NHC precursor-functionalized KAPs with different crosslinking structures were prepared by Friedel-Crafts alkylation of chloromethyl functionalized NHC precursor with tetramethylbenzene, triphenylmethane, and tetraphenylmethane in 1,2-dichloroethane under the catalysis of anhydrous FeCl₃ in high yields[28], and KAP-Pd-PEPPSI-x (x = 1, 2, 3) were then synthesized by treatment of the above NHC precursor-functionalized KAPs with PdCl₂ in 3-chloropyridine with over 90% yields (Fig. 2a and b)[36]. They exhibited good swelling capabilities in organic solvents, and the swelling degree of KAP-Pd-PEPPSI-1, KAP-Pd-PEPPSI-2, and KAP-Pd-PEPPSI-3 in specific dimethyl formamide (DMF) measured by equilibrium swelling method were 239%, 171%, and 127%, respectively, which are inversely correlated with their degree[37]. The high crosslinking degree enabled by the polymeric skeletons of KAP-Pd-PEPPSI-2 and KAP-Pd-PEPPSI-3 are supposed to provide better molecular fences to stabilize the involved Pd species than KAP-Pd-PEPPSI-1.

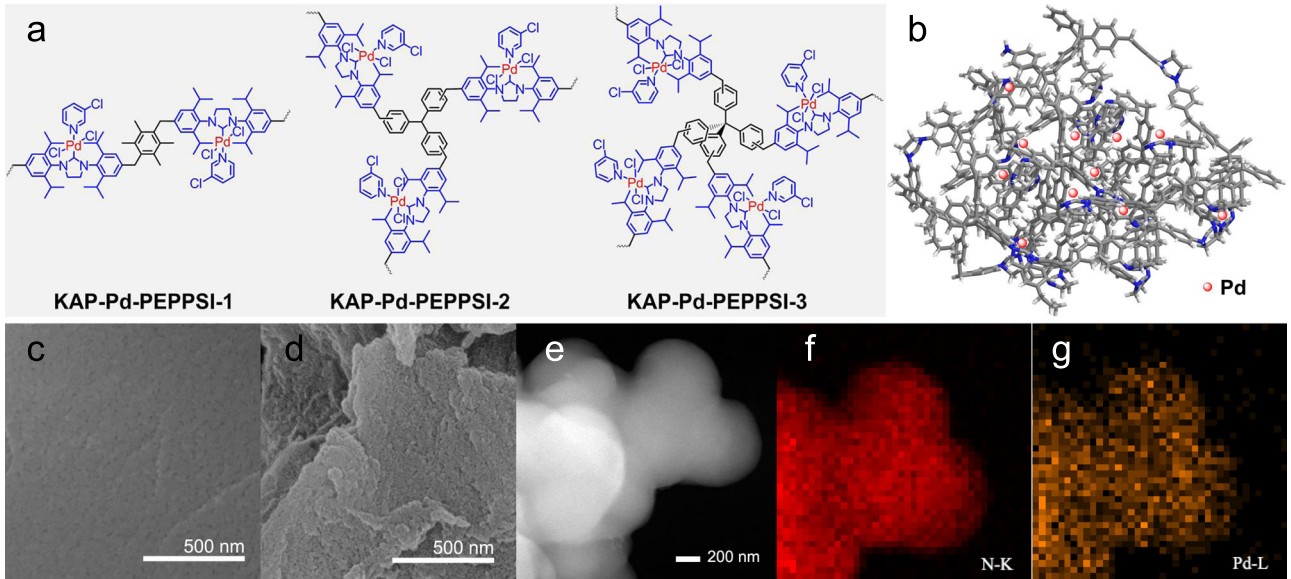

**Fig. 2 Structure and characteristics of KAP-Pd-PEPPSI. a** Structures of KAP-Pd-PEPPSI with different degrees of crosslinking. **b** Simulation model for hypercrosslinked KAP-Pd-PEPPSI-x (x = 2, 3). **c** SEM image of KAP-Pd-PEPPSI-2. **d** SEM image of KAP-Pd-PEPPSI-2 after swelling in DMF for 24 h at 80 °C. **e** STEM image of KAP-Pd-PEPPSI-2. **f**, **g** EDS mapping images for N and Pd in KAP-Pd-PEPPSI-2.

The structural properties of KAP-Pd-PEPPSI were characterized using various spectroscopic measurements. The fourier transform infrared (FT-IR) spectra show bands at 1667, 1465, and 1447 cm$^{-1}$ attributed to the benzene skeleton vibrations (Supplementary Fig. 1)[28]. The peaks at 2860–2970 cm$^{-1}$ are assignable to the alkyl C–H stretch vibration of NHC groups, indicating their incorporation into polymer skeleton[38]. The chemical shifts of $^{13}$C cross polarization magic angle spinning (CP/MAS) nuclear magnetic resonance (NMR) spectroscopy located at 25, 29, 49, 53, and 66 ppm are ascribed to aliphatic methyl, isopropyl, benzyl, imidazolinyl and the quaternary carbon core of tetraphenylmethane (Supplementary Fig. 2), respectively[39], and the broad peaks in the range of 120–160 ppm reveal the presence of aromatic subunits within the polymer networks[40]. These results confirm the integrated structure of NHC-based KAPs. Moreover, the Pd contents determined by inductively coupled plasma mass spectrometry (ICP-MS) in KAP-Pd-PEPPSI-1, KAP-Pd-PEPPSI-2, and KAP-Pd-PEPPSI-3 are 0.24, 0.23, and 0.28 mmol g$^{-1}$, respectively. The Pd 3$d$ X-ray photoelectron spectroscopy (XPS) spectra of KAP-Pd-PEPPSI show peaks at 337.56 (Pd 3$d_{5/2}$) and 342.81 eV (Pd 3$d_{3/2}$), indicating that the Pd in KAP-Pd-PEPPSI are present in +2 oxidation state and strong electron donation from NHC to the Pd(II) exists (Supplementary Fig. 3)[41].

Scanning electron microscopy (SEM) images show that all the KAP-Pd-PEPPSI are consisted of large irregular particles on the order of a few microns with a relatively smooth surface (Fig. 2c and Supplementary Fig. 4a–c), and no metal or salt aggregates are found on the catalyst surface. Besides, the scanning transmission electron microscopy (STEM) images and energy disperse spectroscopy mappings of KAP-Pd-PEPPSI reveal that the Pd and N components are uniformly distributed throughout the catalyst (Fig. 2e–g and Supplementary Fig. 5). It is interesting to find that the porosities of KAP-Pd-PEPPSI are highly dependent on their crosslinking degrees. As seen in Supplementary Fig. 6, the Brunauer-Emmett-Teller (BET) surface area in KAP-Pd-PEPPSI-1 is 227 cm$^3$ g$^{-1}$, which decreases to 10 cm$^3$ g$^{-1}$ and 7 cm$^3$ g$^{-1}$, respectively, in KAP-Pd-PEPPSI-2 and KAP-Pd-PEPPSI-3 with higher crosslinking degree. Although the hypercrosslinking structure results in the decrease in the number of intrinsic pores, the swelling properties of KAP-Pd-PEPPSI in solid–liquid condition benefit the access of reactants to active sites[42], which can be illustrated in Fig. 2d and Supplementary Fig. 4d–f that a large number of small cracks are formed and the surface roughness of these particles increases markedly after swelling in DMF. In this case, the unique elimination of the large intrinsic pores enabled by hypercrosslinked polymer skeleton and the controlled swelling properties of KAP-Pd-PEPPSI may produce an enhanced spatial confinement effect, which is believed to be crucial for stabilizing the active Pd species with synergetic coordination with NHC sites[43,44].

To systematically understand the confinement and coordination effects on the stability of KAP-Pd-PEPPSI catalysts, other supported Pd metal catalysts with or without N, P heteroatom doping, such as Pd@C, N-doped carbon supported Pd nanoparticles (Pd@NC)[45], P-doped carbon supported Pd nanoparticles (Pd@PC)[46], porous silica supported Pd nanoparticles (Pd@SiO$_2$)[47], as well as supported Pd complex precatalysts with phosphine and NHC as coordinative site, such as PPh$_3$-functionalized KAP supported Pd(II) complex (KAPs-Pd-PPh$_3$) and porous organic polymer with built-in Pd(II)-NHC complex (POPs-Pd-PEPPSI)[28,48], have been prepared according to the reported procedures, and the detailed synthetic processes are given in the attached methods and supplementary information (Supplementary Fig. 7).

**Fluorocarbonylation of Indole.** Acid halides are an important class of carbonyl compounds which are widely used as acyl reagents or intermediates to produce numerous carbonyl compounds[49]. Conventional synthesis of acid halides from transformations of C–H bonds usually undergo carboxylation or oxidation to generate the carboxylic acids, and then react with halogenating reagents, such as PCl$_3$, thionyl or oxalyl chloride, to produce the corresponding acyl chlorides, which generate a large amount of wastes[50]. Note that Pd-catalyzed carbonylations of aryl halides have been developed to be facile alternatives to synthesize acid halides[51–53]. Particularly, Andersen et al. achieved an unprecedented breakthrough in establishing carbonylation of aryl halides to produce acid chlorides via sterically encumbered $^t$Bu$_3$P and CO coordination enabled reductive elimination of ($^t$Bu$_3$P)

**Table 1 Catalytic performances of supported Pd catalysts[a].**

| Entry | Catalyst | $S_{BET}$ (m²g⁻¹) | Yield[b] (%) | Pd leaching[c] (%) |
|---|---|---|---|---|
| 1 | Pd-PEPPSI | | 69 | |
| 2 | Pd@C | 619 | 93 | 2.2 |
| 3 | Pd@NC | 478 | 91 | 1.6 |
| 4 | Pd@PC | 557 | 87 | 1.3 |
| 5 | Pd@SiO₂ | 482 | 82 | 1.0 |
| 6 | KAP-Pd-PPh₃ | 473 | 91 | 0.8 |
| 7 | POP-Pd-PEPPSI | 458 | 90 | 0.5 |
| 8 | KAP-Pd-PEPPSI-1 | 227 | 89 | 0.5 |
| 9 | KAP-Pd-PEPPSI-2 | 10 | 89 | 0.4 |
| 10 | KAP-Pd-PEPPSI-3 | 7 | 87 | 0.2 |

[a]Reaction conditions: **1a** (0.25 mmol), CsF (1 mmol), Pd catalyst (5 mol%), I₂ (0.5 mmol), base (0.5 mmol), DMF (2 mL), 80 °C, 24 h, 2.0 MPa.
[b]HPLC yield.
[c]Pd content in the solution after completion of the reaction.

(CO)Pd(COAr)Cl[51]. On the other hand, direct C–H functionalization has greatly boosted the carbonylation methods in direct preparation of carboxylic compounds without using prefunctionalized substrates like aryl halides[54]. However, to the best of our knowledge, carbonylation of C–H bonds to acid halides has not been reported. Compared with the acid chlorides (C-Cl bonds, ~78 kcal/mol), acid fluorides are more thermodynamically stable (C-F bonds, ~105 kcal/mol) and can be readily handled for further transformation under simple conditions[55]. We herein select carbonylation of indole to corresponding acid fluoride as model reaction to investigate the catalytic activity and stability of KAP-Pd-PEPPSI, and this one-pot transformation is designed to proceed via oxidative iodination of C–H bond first and then carbonylation.

Initial screening of reaction conditions was conducted with Pd-catalyzed fluorocarbonylation of 1-methyl-1H-indole (**1a**) using KAP-Pd-PEPPSI-2 as the precatalyst (Supplementary Table 2). After a series of primary investigations, the yield of indole-3-carbonyl fluoride (**A**) reached 89% when using CsF as the fluorine source and DABCO as the base (Supplementary Table 2, entry 5). 3 mol% of KAP-Pd-PEPPSI-2 produced 88% yield of **A**, and 83% yield of **A** was obtained when lowering the catalyst amount to 2 mol% (Supplementary Table 2, entry 12 and 13). Further reducing the catalyst loading to 1 mol% gave **A** in 54% yield, which could be promoted to 86% upon extending reaction time to 48 h (Supplementary Table 2, entry 14 and 15). Under the optimized reaction conditions as presented in Table 1, all Pd-based heterogeneous catalysts provided **A** in high yields (82–93%, entries 2–10), indicating for the first time the availability of synthesizing acid halides by heterogeneous carbonylation. Surprisingly, a relatively lower yield (69%) was obtained when using molecular Pd-PEPPSI as homogeneous control catalyst (entry 1), and the formation of Pd black in the solution was observed (Supplementary Fig. 8a), suggesting the significant challenge of this reaction condition to catalyst stability. Notably, the Pd contents in reaction solutions were tested by ICP-MS to be 5.5–91% of the total Pd when treating different supported Pd catalysts with the initial reactants for 4 h in the absence of CsF (Washing Test) or keeping yield of **A** at low levels (Supplementary Table 3)[14]; however, only a very small portion of Pd remained in solution after the completion of the reactions

(Table 1). These results clearly demonstrate the "Cocktail" type nature of supported Pd-catalyzed carbonylation reactions, in which a Pd dissolution and redeposition process is involved[12]. The STEM morphologies of used catalysts revealed that severe Pd aggregation and uneven dispersion were detected on Pd@C, Pd@NC and Pd@SiO₂ (Supplementary Fig. 8b, c and e). Notably, the hierarchical porous Pd@PC with P-doping was found to be more stable than other supported Pd catalysts, but the formation of Pd aggregates was still not avoided (Supplementary Fig. 8d). Gratifyingly, the organic polymer supported Pd complex catalysts exhibited much better stabilities than the Pd catalysts over inorganic supports. As shown in Supplementary Fig. 9, no obvious changes compared to the fresh catalysts were observed in the STEM images of KAP-Pd-PPh₃, POP-Pd-PEPPSI, and KAP-Pd-PEPPSI after the completion of the reaction, and the Pd components were homogeneously distributed throughout these catalysts without formation of detectable nanoparticles or aggregates. It can be primarily deduced that the polymer supports with built-in electron-donating P- or NHC-ligands can prevent the in situ formed active Pd species from aggregation in this Pd dissolution and redeposition process.

Then, we selected KAP-Pd-PPh₃, POP-Pd-PEPPSI, and KAP-Pd-PEPPSI-2, which have similar skeletal structures to study their recyclabilities in carbonylation reaction. As shown in Fig. 3a, all these catalysts could be used at least four times without a significant decline in the activity. However, gradual activity decreases were observed in KAP-Pd-PPh₃ and POP-Pd-PEPPSI when further increasing the reaction cycles, and obvious Pd particle size changes evidenced the deactivation of these two catalysts via metal aggregation (Fig. 3c and Supplementary Fig. 10). To our delight, KAP-Pd-PEPPSI-2 could be recycled for nine times without significant loss of activity (Fig. 3a). TEM images of spent KAP-Pd-PEPPSI-2 demonstrated that the catalyst after the 5th use was very similar to the fresh one (Supplementary Fig. 11). It retained a good dispersion of Pd and no large-sized nanoparticles or aggregates were observed (Fig. 3c and Supplementary Fig. 12). Moreover, the composition and chemical state of the surface Pd on the above catalysts after the 5th cycle were investigated by XPS, revealing that there exist three types of Pd species with Pd(0) and Pd(II) oxidation states on the recycled catalysts generated from KAP-Pd-PPh₃, POP-Pd-PEPPSI and

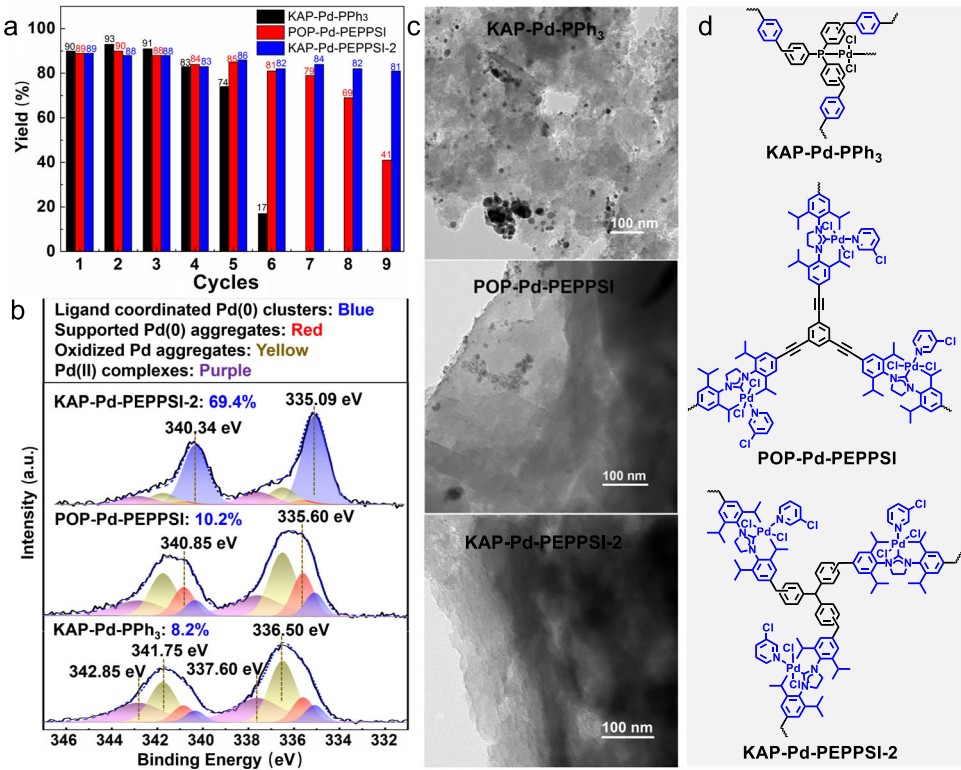

**Fig. 3 Recyclability of the selected catalyst. a** Recyclability of the supported Pd complex precatalysts in fluorocarbonylation of indole. Reaction conditions: **1a** (0.25 mmol), CsF (1 mmol), catalyst (5 mol%), $I_2$ (0.5 mmol), base (0.5 mmol), DMF (2 mL), 80 °C, 24 h, 2.0 MPa. **b** Pd XPS spectra and **c** TEM images of different catalysts after being used for five times. **d** Structures of the catalysts used here.

KAP-Pd-PEPPSI-2 (Supplementary Fig. 13). The content of the oxidized Pd(II) species is 79.6% (yellow and purple areas) on the supported catalyst originated from KAP-Pd-PPh₃, while only about 8.2% of Pd(0) clusters, which are electronically coordinated by phosphine ligand, and 13.3% of Pd(0) aggregates supported by the polymer skeleton, are observed with their Pd $3d_{5/2}$ core level peaks located at 335.09 eV and 335.60 eV, respectively[56]. Whereas, the supported Pd catalyst from KAP-Pd-PEPPSI-2 bearing similar polymer skeleton to KAP-Pd-PPh₃ has different Pd composition, revealing high content of Pd(0) clusters (69.4%, blue areas) coordinating with the NHC sites[57]. These results suggest that the NHC site demonstrates better electron-donating effect, compared with phosphine, in stabilizing Pd species during their dissolution into reaction solution and redeposition onto the hypercrosslinked polymeric catalyst. Another POP-Pd-PEPPSI with the same built-in NHC sites but differently structured polymer skeleton, showed much lower content (10.2%, blue curves) of the NHC-coordinated Pd(0) clusters, indicating that the polymeric skeleton also plays a very important role on stabilizing Pd species. It can be preliminarily inferred from above results that the hypercrosslinked KAP with built-in NHC sites acts as ideal molecular fence, wherein the polymer skeleton and the strong electron-donating NHC work synergistically to suppress the Pd aggregation in the Pd dissolution and redeposition process.

**Investigation on stabilizing effect.** Following the above basic understanding, the coordination influence of NHC site was investigated by using two structurally similar KAP supports, KAP-4 and KAP-2 with or without incorporated NHC precursor as shown in Fig. 4a, to interact with the in situ formed Pd species from PdCl₂ in the model fluorocarbonylation of indole. It was found that, although the PdCl₂/KAP-4 catalytic system gave a

83% yield of acid fluoride **A** in the fresh reaction, only trace amount of the product was obtained in its second use, which could be explained by its TEM images showing the formation of large aggregates with size up to hundreds of nanometers on the catalyst surface (Fig. 4c and Supplementary Fig. 14) and the Pd $3d$ XPS spectra which support the formation of Pd(0) aggregates with Pd $3d_{5/2}$ core level peaks at 335.60 eV (Fig. 4b). These results suggest a poor metal-support interaction between KAP-4 and dissolved Pd species, and it also can be experimentally deduced from its second use that this types of electron-deficient Pd(0) aggregates show very low activity toward this one-pot carbonylation of indole. On the contrary, the PdCl₂/KAP-2 catalytic system provided a 73% yield of **A** in its first use, and a higher **A** yield (89%) was obtained in the second reaction cycle. The corresponding TEM images revealed a good dispersion of Pd components and no obvious aggregates were detected (Fig. 4c and Supplementary Fig. 15). Besides, the deconvoluted Pd $3d$ peaks at 335.09 eV and 340.34 eV confirmed the formation of NHC-coordinated Pd(0) clusters (Fig. 4b). These findings indicate that the electron-rich NHC sites generated via deprotonation of the NHC precursor in KAP-2 can electronically donate to and then effectively stabilize the dissolved Pd species via coordination interaction in the reaction process.

Having figured out the electronic coordination effect of NHC sites, we next investigated the influence of the catalyst structure on the stability of active Pd species. For this purpose, three KAP-Pd-PEPPSI-x (x = 1, 2, 3) precatalysts with different crosslinking degrees (Fig. 1a) were used to investigate their recyclabilities in carbonylation of indole at about 50% yield of acid fluoride **A**. As shown in Fig. 4d, a gradual loss of activity was observed in low-crosslinking KAP-Pd-PEPPSI-1 with a chain structure, and the TEM images of the spent catalyst after the 6th cycle showed large amount of irregular aggregates (Fig. 4e and

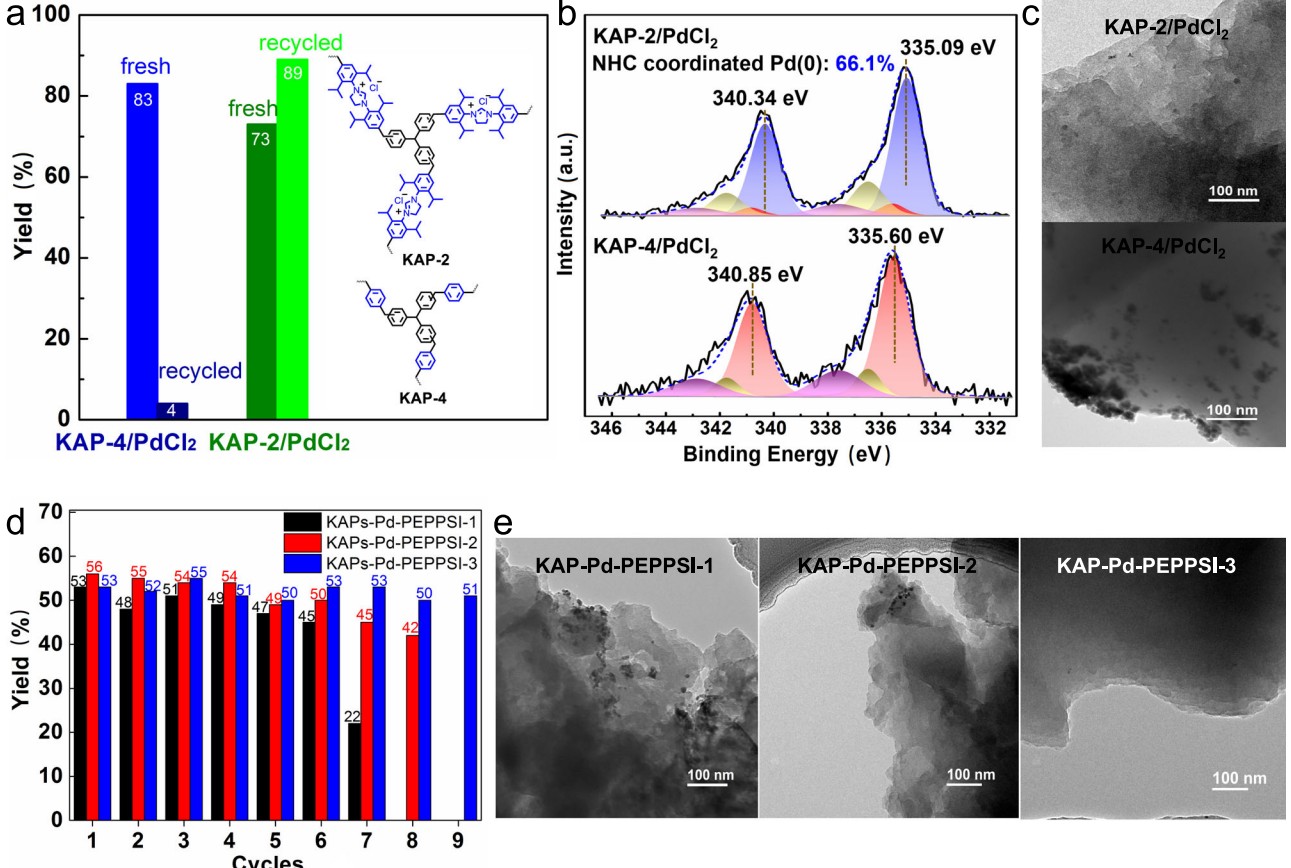

**Fig. 4 Stabilizing effect of KAP. a** Control experiments to determine the stabilizing effect of NHC under the optimized reaction conditions. **b** Pd XPS spectra and **c** TEM images of recycled KAP-2/PdCl₂ and KAP-4/PdCl₂. **d** Control experiments to determine the stabilizing effect of catalyst structure. Reaction conditions: **1a** (0.25 mmol), CsF (1 mmol), Pd catalyst (2.5 mol%), I₂ (0.3 mmol), base (0.5 mmol), DMF (2 mL), 80 °C, 16 h, 2.0 MPa. **e** TEM images of different KAP-Pd-PEPPSI-x after the 6th cycle.

Supplementary Fig. 16), indicating a severe particle aggregation on the catalyst surface. KAP-Pd-PEPPSI-2 with its NHC moiety linked by triphenylmethane group, which has a higher cross-linking degree than that of KAP-Pd-PEPPSI-1, delivered an enhanced catalytic stability. A small amount of aggregates with size less than twenty nanometers were observed in the TEM images of the spent KAP-Pd-PEPPSI-2 (Fig. 4e and Supplementary Fig. 16). KAP-Pd-PEPPSI-3 which has the highest cross-linking degree among these three catalysts, showed even better stability than KAP-Pd-PEPPSI-2. The catalyst could be used at least nine times without loss of activity and no any observable aggregates were detected on the TEM images of used catalyst (Fig. 4e and Supplementary Fig. 16). The above results indicate that the highly hypercrosslinking moieties of the KAP skeleton work like a molecular fence to confine the in situ formed active Pd species inside each polymeric domain (Supplementary Fig. 17 and 18), which benefits the recapture of the dissolved Pd species by the support via electronic interaction/coordination and prevents the supported small Pd clusters from aggregation on the support surface[28,58,59]. Therefore, it can be concluded from these two parts of studies that the synergetic cooperation of the polymeric confinement and the electronic NHC coordination inside the hypercrosslinked KAP skeleton contributes to the efficient Pd stabilization during their dissolution into the reaction mixture and redeposition back onto the KAP catalyst.

**Investigation on carbonylation process**. To get more insights into this KAPs-supported Pd-catalyzed carbonylative C–C bond

formation reaction under solid–liquid condition, the time-course analyses of Pd evolution and distribution as well as corresponding reaction activity in fluorocarbonylation of indole were carried out using typical Pd@C and the optimized KAP-Pd-PEPPSI-3 as the catalyst, respectively. As revealed in recent studies that the molecular Pd/NHC complexes may undergo a dynamic process during the catalysis by Pd-C_{NHC} bond cleavage to produce either NHC-coordinated or ligandless active Pd species[60–62]. Here, the KAP-Pd-PEPPSI-3 experienced a long activation time to reduce the polymeric Pd(II)-NHC precatalyst to the active NHC-coordinated Pd(0) clusters, the formation of which could be testified by the complete prohibition of the reaction after adding mercury (mercury poisoning experiment)[9,63], and **A** started to be generated with its yield rapidly increasing to 87% in the next six hours (blue curve with dots). On the other hand, the concentration of Pd species in solution slowly increased in the first four hours, and it reached a maximum of 5.5% of total Pd when A started to be generated and then rapidly decreased (blue curve with squares). Such an obvious mismatch between the Pd content in solution and the product yield indicates a fast redeposition of the regenerated Pd(0) onto the polymeric support. Therefore, the Pd species involved in the carbonylation were kept recycling via following the process of initial detachment from the polymer catalyst into solution, catalyzing carbonylation and redeposition back onto the polymer without deactivation, and thus the very active Pd species avoided accumulating in the solution and were maintained at a dilute concentration, which is significantly helpful to suppress Pd aggregation[64]. Conversely, the typical Pd/C

displayed a different Pd distribution, the Pd content in solution increased rapidly to about 37.3% of total Pd in about two hours after the reaction started (purple curve with squares), and then remained at a high concentration level (28.8–40.7%) for over six hours with the yield of **A** increasing to about 80% (purple curve with dots in Fig. 5a and Supplementary Table 4). Although most of the dissolved Pd (only 2.2% of total Pd remained in solution as present in Table 1) redeposited back onto the carbon support, many of them have grown into bigger sized Pd aggregates with ununiform dispersion as evidenced by STEM (Supplementary Fig. 8b). Compared to the conventional supported catalysts, this polymeric catalyst with swelling property could efficiently trap the active Pd species inside the skeletal domain by the confinement of the polymeric skeleton and coordination of the NHC sites, keeping the dissolved Pd species at a low concentration level in solution. Moreover, the hot filtration test of the KAP-Pd-PEPPSI-3 catalyst at its peak Pd concentration in solution only gave a 30% yield of **A** after the filtrate reacted for 24 h (red line in Fig. 5b), and Pd black was clearly observed, indicating a fast deactivation of the dissolved Pd without the stabilization of polymer skeleton and NHC.

The stabilizing effect based on the above experimental results is proposed (Fig. 5c). First, the built-in polymeric Pd(II) complex is in situ reduced to NHC-coordinated Pd(0) clusters, wherein the NHC group acts as an electron-donating ligand that increases the electron density of Pd(0), thereby stabilizing the Pd(0) clusters from aggregation[65]. Next, the oxidative addition of Pd(0) species into the C-I bond of 3-iodo-1-methylindole, generated by iodination of C3–H bond of indole, leads to the detachment of Pd from the clusters, affording the representative Pd(II) intermediate [Indole]PdI confined inside each of the hypercrosslinked polymer domains[13]. Subsequently, CO insertion, fluoride exchange and reductive elimination give product **A** with regeneration of electron-deficient Pd(0) species, which are then recaptured onto the polymeric skeleton mainly by electronic interaction with NHC sites to recover the supported Pd(0) clusters for next reaction cycle. As a result, the hypercrosslinked KAPs with built-in NHC sites function as molecular fences to effectively trap the active Pd species inside the polymer during the reaction process and reload them after completion of one catalytic cycle, which prevent the Pd species from accumulating in solution and the formation of large inactive aggregates.

**Applications in other C–C bond formations.** Besides above carbonylative synthesis of acid fluoride, carbonylations of aryl halides with hydrogen, alcohol, amine, organometallic compound, have been extensively used to produce aldehyde, ester, amide, and ketone[34]. However, some unconventional nucleophiles, such as weakly nucleophilic aromatic amines (especially NH-heteroarenes)[66], strongly coordinating CN−[67] and intolerable N₃⁻[68] have been scarcely used for carbonylative transformation due to their inherent limitations, and corresponding heterogeneous carbonylation procedures have not been reported yet. But nevertheless, the corresponding carbonyl compounds of above nucleophiles are important building blocks in the syntheses of fine chemicals and bioactive compounds[69–71]. Based on the establishment of KAP-Pd-PEPPSI catalyzed direct fluorocarbonylation to access acid fluoride, we subsequently investigated carbonylations of indole with above three challenging nucleophiles to synthesize indole-derived acylamides, aroyl azides and aroyl cyanides in one-pot cascade or sequential manner, respectively.

We first tried to develop a direct aminocarbonylation of indole with weakly nucleophilic N–H of pyrazole via the combination of fluorocarbonylation and subsequent acylation in a cascade

reaction manner without isolation of acid fluoride intermediate in a single reactor. It is necessary to note that the conventional direct aminocarbonylation of C–H bond with N–H moiety took selectivity challenge, because it needs an oxidative condition which inevitably leads to the oxidative carbonylation of N–H bonds, giving undesirable urea. Herein, initial screenings indicated that the Arndtsen's chlorocarbonylation methodology with tetrabutyl ammonia chloride as Cl⁻ resource by specific Pd (PtBu₃)₂ catalyst, and Manabe's fluorocarbonylation procedure with KF as F⁻ reagent by Pd(OAc)₂/xantphos catalyst, only provided 14% and 11% yield of product **3aa**, respectively, (Supplementary Table 5), suggesting that these two catalytic procedures afforded low yields of corresponding acid chloride and acid fluoride from carbonylation of in situ formed iodoindole. Whereas, a 74% yield of **3aa** was obtained upon using our KAP-Pd-PEPPSI-3/CsF catalytic system (Table 2 and Supplementary Table 5), and no urea byproduct was observed. The corresponding control experiments also revealed that the in situ generated A could readily react with pyrazole (**2a**) to give the corresponding indole-3-amide, while no fluorocarbonylation or aminocarbonylation occurred in the absence of fluoride or replacing fluoride with chloride, bromide or iodide (Supplementary Table 6).

Under the optimal reaction conditions, the substrate scope was explored. The aminocarbonylations of N-methyl indoles, bearing a substituent such as -Me, -Cl, -Br, and -OMe on the benzene ring, all proceeded smoothly to produce the corresponding indole-3-amides (**3ba-3ia**) in 57–91% yields. The reaction of pharmaceutically interesting substrate, 1-methyl-1H-pyrrolo[2,3-b]pyridine[72], gave **3ja** in 70% yield. Indoles with different N-protective groups were also well-tolerated, both N-allyl indole and N-benzylindole showed good reactivity in the one-pot synthesis of **3ka** and **3ma**. The cascade reaction of N-methyl pyrrole instead of indole also took place efficiently, selectively giving monocarbonylated **3la** in 62% yield. Other substituted pyrazoles, imidazole, benzimidazole and aniline derivatives were also good nucleophiles for this carbonylation, providing **3ab-3aj** in 59–85% yield. Importantly, KAP-Pd-PEPPSI-3 demonstrated outstanding recyclability in this cascade process under more complicated conditions than that of fluorocarbonylation, and could be used at least five runs without any loss of activity in the synthesis of **3aa** (77%).

Carbonylation of the nonprefunctionalized substrate with N₃⁻ as the nucleophile (azidocarbonylation) provides a direct access to versatile aroyl azide, wherein two challenges exist: (1) the aroyl azide product is prone to react with commonly used tertiary phosphine ligand in carbonylation reactions to afford the corresponding benzoyl azaylide (known as the Staudinger reaction)[73]; (2) the N₃⁻ coordinated to Pd can be converted into NCO⁻ in the presence of CO even under mild reaction conditions[74]. Taking advantage of the present phosphine-free KAP-Pd-PEPPSI catalytic system, we tried a one-pot and two-step procedure to establish this azidocarbonylation transformation, first fluorocarbonylation of indole to afford acid fluoride and then discharging excess CO for azidonation with NaN₃. Following this sequential procedure (Table 3), azidocarbonylation of N-methyl indoles bearing different substituents provided **4a-4i** in 49–92% yield. The reaction of 1-methyl-7-azaindole with a pyridine moiety also proceeded smoothly, giving a 69% yield of **4j**. The investigations on the substituent effect revealed that the electron-donating groups are more favorable for this carbonylation (**4k-4n**). When 1-benzyl-1H-indole was employed, **4o** was obtained in 81% yield. Besides, N–H indole could also furnish the corresponding product in 64% yield (**4p**) by prolonging the reaction time to thirty hours. Last but not the least, KAP-Pd-PEPPSI-3 could be used at least five runs without loss of its

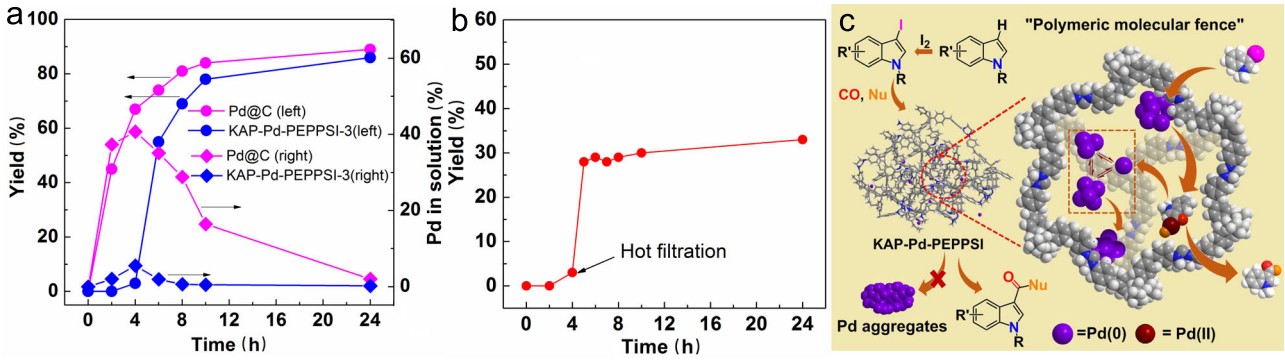

**Fig. 5 Reaction process investigation. a** Kinetic investigation. **b** Hot filtration test. Under the optimized reaction conditions, the reaction mixture was quickly divided into two parts by centrifugation after reaction for 4 h, and the upper clear liquid proceeded to react for another 20 h in the presence of CsF. **c** Stabilizing effect of the hypercrosslinked polymer on active Pd species in KAP-Pd-PEPPSI catalyzed fluorocarbonylation of indole.

---

**Table 2 Aminocarbonylation of indoles and their analogues with aromatic amines[a].**

R[2]–indole (X = C or N) **1** + NH–Ar **2** → Cat. (5 mol%), CsF, I$_2$, DABCO, CO (2.0 MPa), 80 °C → **3**

X = C or N **1**

R[2] = H, **3aa**, 74% (77% 5[th])
R[2] = 4-Me, **3ba**, 83%
R[2] = 5-Me, **3ca**, 81%
R[2] = 6-Me, **3da**, 91%
R[2] = 7-Me, **3ea**, 69%
R[2] = 2-Me, **3fa**, 57%

**3ga**, 69%

**3ha**, 64%

**3ia**, 73%

**3ja**, 70%

**3ka**, 56%

**3la**, 56%

**3ma**, 62%

R[3] = 3-Me, **3ab**, 72%
R[3] = 3,5-Me, **3ac**, 63%

**3ad**, 85%

**3ae**, 68%

**3af**, 71%

**3ag**, 67%

R[4] = H, **3ah**, 64%
R[4] = 4-OMe, **3ai**, 70%
R[4] = 4-Cl, **3aj**, 59%

[a]Reaction conditions: KAP-Pd-PEPPSI-3 (5 mol%), indole (0.25 mmol), amine (1.25 mmol), I$_2$ (0.5 mmol), DABCO (0.5 mmol), CsF (1.0 mmol), CO (2.0 MPa), DMF (2.0 mL), 80 °C, 24 h, isolated yield.

**Table 3 Azidocarbonylation of indoles and their analogues to synthesize indole-3-carbonyl azides[a].**

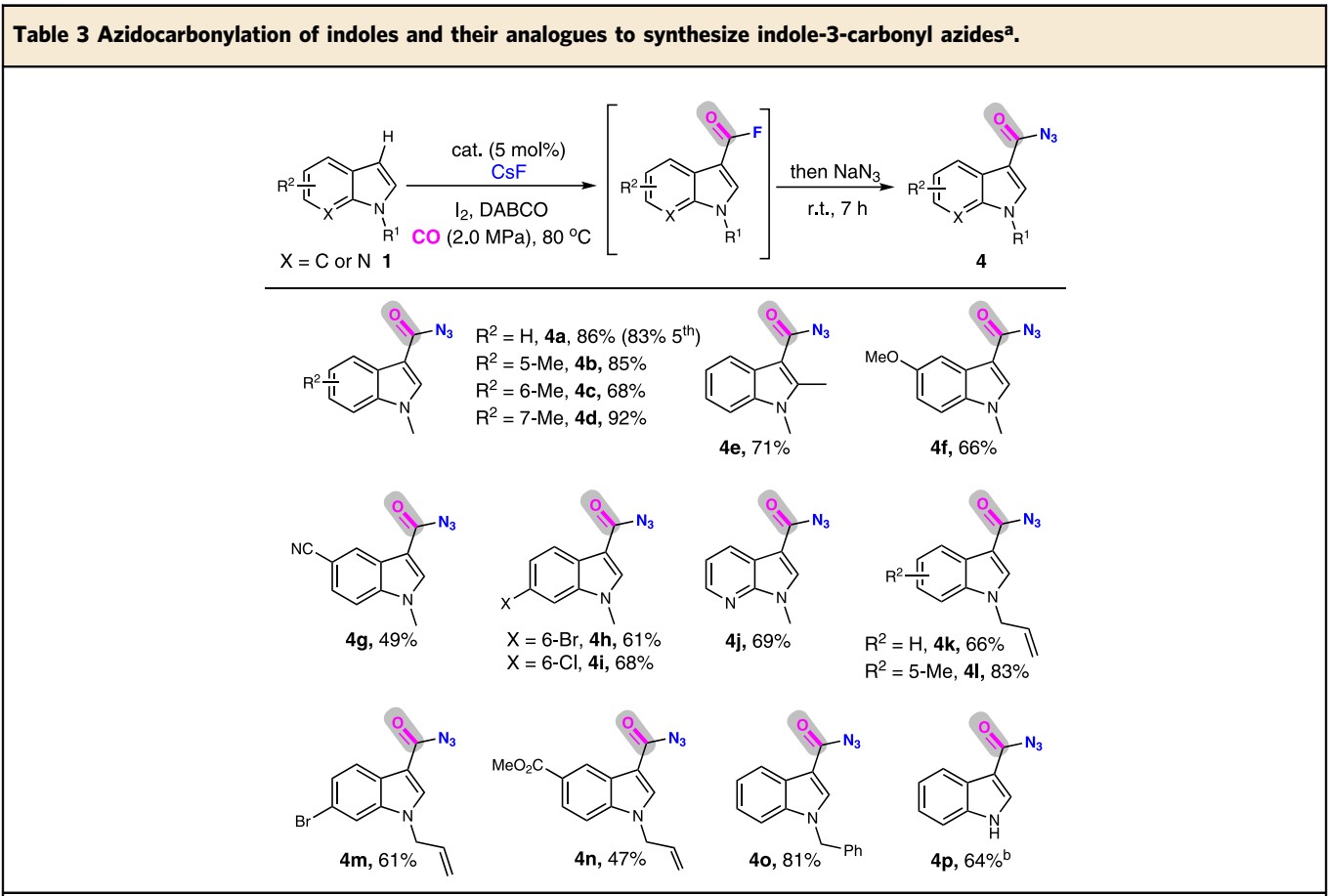

R² = H, **4a**, 86% (83% 5th)
R² = 5-Me, **4b**, 85%
R² = 6-Me, **4c**, 68%
R² = 7-Me, **4d**, 92%

**4e**, 71%

**4f**, 66%

**4g**, 49%

X = 6-Br, **4h**, 61%
X = 6-Cl, **4i**, 68%

**4j**, 69%

R² = H, **4k**, 66%
R² = 5-Me, **4l**, 83%

**4m**, 61%

**4n**, 47%

**4o**, 81%

**4p**, 64%[b]

[a]Reaction conditions: KAP-Pd-PEPPSI-3 (5 mol%), indole (0.25 mmol), I₂ (0.5 mmol), DABCO (0.5 mmol), CsF (1.0 mmol), DMF (2.0 mL), CO (2.0 MPa), 80 °C, 24 h. Then NaN₃ (1.0 mmol) was added, room temperature, 7 h, isolated yield.
[b]30 h.

catalytic activity (83% yield), and the easily formed isocyanate or other byproducts were not detected by this one-pot sequential procedure.

Carbonylation with cyanide anion as the nucleophile (cyanocarbonylation) could simultaneously introduce two functional groups (C=O and C≡N) for diverse transformations, and the challenges therein are the poisoning of the catalyst by strong coordination of CN⁻ to Pd and byproduct formation via coupling between ArX with CN⁻ without CO incorporation. Following the above one-pot sequential procedure, we have established a cyanocarbonylation of indole to prepare corresponding indole-3-carbonyl nitrile. As shown in Table 4, this protocol has good tolerance to diverse functional groups at different positions of N-methylindole, giving the corresponding indole-3-carbonyl nitriles (**5a-5i**) in good to excellent yields (60–85%). 1-Methyl-7-azaindole was also a good substrate for this carbonylation, giving **5j** in 55% yield. The reaction of various substituted N-allyl indole also proceeded smoothly to afford the desired **5k-5m** in 57–75% yields. Using pharmaceutically interesting N-benzylindole as the substrate[75], **5n** was obtained in 77% yield. Besides, N–H indole also gave the corresponding acylcyanide **5o** in good yield. Moreover, KAP-Pd-PEPPSI-3 displayed remarkable reusability and provided a high yield of **5a** in its 5th use.

To illustrate the useful application of KAP-Pd-PEPPSI catalyzed carbonylations of indole in synthetic community, we performed the synthesis of 1-benzyl-N-(4-chlorophenyl)-1H-indole-3-carboxamide **6**[76], 4,5-bisindolylpyrazolone **7c**[77] and nortopsentin analogue **8c**[78], which show potent biological activities as antioxidant, glycogen synthase kinase inhibitor and

antiviral, using the developed aminocarbonylation, azidocarbonylation, and cyanocarbonylaiton as the key synthetic tools. As shown in Fig. 6, aminocarbonylation of N-benzylindole with 4-chloroaniline via a one-pot two-step procedure generated **6** with a satisfactory yield (72%). Nucleophilic addition of indole-3-carbohydrazide to 3-isocyanato-N-methylindole **7a** that was obtained by one-pot azidocarbonylation of N-methylindole and then refluxing in toluene, gave bisindole carbohydrazide **7b** in an overall yield of 81%, further cyclization of **7b** afforded **7c** in 76% yield. As to the application of indole-3-carbonyl cyanide generated by cyanocrbonylation of indole, nucleophilic substitution of **5o** with oxotryptamine **8a**, which was synthesized from hydrogenation of another molecule of **5o**, produced 98% yield of indole-3-carboxamide **8b**, and its subsequent cyclization delivered **8c** in 81% yield. These three representative syntheses demonstrate the outstanding potential of present KAP-Pd-PEPPSI catalyzed one-pot carbonylation in pharmaceutical synthesis.

Besides efficient and stable applications in above carbonylative C–C bond formations, the optimized KAP-Pd-PEPPSI-3 also displayed excellent recyclability with up to 21 cycles and >95% product yield in another representative Suzuki-Miyaura coupling reaction at low catalyst loading (0.4 mol%) (Supplementary Fig. 19). STEM images of KAP-Pd-PEPPSI-3 after the 21st cycle showed uniformly distributed nanoparticles without formation of large aggregates (Supplementary Fig. 20c and d). XPS results revealed the presence of a strong metal-ligand interaction between Pd nanoparticles and NHC groups (Supplementary Fig. 21). On the contrary, typical Pd@C displayed a significant

**Table 4 Cyanocarbonylation of indoles and their analogues to synthesize indole-3-carbonyl cyanides[a].**

$R^2$ = H, **5a**, 82% (79% 5th)
$R^2$ = 4-Me, **5b**, 81%
$R^2$ = 5-Me, **5c**, 85%
$R^2$ = 6-Me, **5d,** 71%
$R^2$ = 7-Me, **5e**, 83%

**5f**, 60%

**5g**, 79%

**5h**, 63%

**5i**, 71%

**5j**, 55%

**5k**, 67%

**5l**, 75%

**5m**, 57%

**5n**, 77%

**5o**, 59%[b]

[a]Reaction conditions: KAP-Pd-PEPPSI-3 (5 mol%), indole (0.25 mmol), $I_2$ (0.5 mmol), DABCO (0.5 mmol), CsF (1.0 mmol), DMF (2.0 mL), CO (2.0 MPa), 80 °C, 24 h. Then TMSCN (1.0 mmol) and CsF (1.0 mmol) were added, room temperature, 7 h, isolated yield.
[b]30 h.

decrease in activity after the 6th cycle and the formation of large particle aggregates (up to 500 nm in size) suggests the severe Pd aggregation during the reaction (Supplementary Fig. 20a and b). Interestingly, 0.4 mol% of KAP-Pd-PEPPSI-3 could also effectively catalyze the cross-coupling of representative unactivated phenyl chlorides and N-heterocyclic chlorides as well as the challenging benzyl chloride with phenylboronic acid into the corresponding biphenyls in excellent yields (92–98%, Supplementary Table 7). These results further confirm the super stability of hypercrosslinked KAP-Pd-PEPPSI-3 and highlight its potential applications in other C–C bond formation reactions.

## Discussion
In summary, we have developed a series of efficient and highly stable supported Pd catalysts for different C–C bond formations under solid–liquid catalysis. For the first time, the synthesis of acid halide by heterogeneous carbonylation was established via KAP-Pd-PEPPSI catalyzed fluorocarbonylation of indole with enhanced catalytic stability surpassing other conventional supported Pd catalysts. Investigations on the stabilizing effect revealed that the hypercrosslinked polymer skeleton of the catalyst worked like a molecular fence to effectively trap the dissolved Pd species inside the swollen polymer by the electronic NHC coordination and polymeric confinement in the reaction process, which facilitated the reloading of active Pd species back onto the skeletal support and avoids their accumulation in reaction solution, thus suppressing the catalyst deactivation caused by Pd aggregation. Moreover, the optimized KAP-Pd-PEPPSI-3 also

showed excellent activities and durabilities in other three types of one-pot fluorocarbonylation/acylation reactions, providing effective procedures to access unconventional carbonyl derivatives (e.g., heterocyclic acylamide, aroyl azides, and aroyl cyanides) that are valuable compounds for functional applications and diverse transformations. Likewise, KAP-Pd-PEPPSI-3 also displayed excellent recyclability in representative Suzuki-Miyaura coupling reaction at low catalyst loading. This work provides an effective strategy for the development of metal supported catalysts with high stability, which will also boost their efficient applications in advanced synthesis under solid–liquid catalysis.

## Methods
**Synthesis of KAP-Pd-PEPPSI-1**. 1,3-Bis(2,6-diisopropyl-4-(chloromethyl)phe-nyl)-4,5-dihydro-1H-imidazol-3-iumchloride (524 mg, 1 mmol), anhydrous FeCl₃ (1.95 g, 12 mmol), and 1,2,4,5-tetramethylbenzene (134 mg, 1 mmol) were mixed together in anhydrous 1,2-dicholoroethane. The mixture was refluxed at 80 °C under N₂ for 24 h, and the resultant precipitates were collected by filtration and washed continually with methanol until the filtrate was clear and then treated by Soxhlet extraction in methanol for 24 h. The obtained reddish brown powder was dried under vacuum at 100 °C for 12 h to afford the crosslinked polymer with built-in NHC precursor (KAP-1, 68% yield). Then, a mixture of KAP-1 (384 mg), PdCl₂ (25 mg, 0.14 mmol), and anhydrous K₂CO₃ (345 mg, 2.5 mmol) in 5 mL of 3-chloropyridine was refluxed for 16 h, after which the precipitates were collected by filtration and washed successively with DMF, water and methanol. Further drying under vacuum at 60 °C for 24 h gave the KAP-Pd-PEPPSI-1 in 94% yield.

**Synthesis of KAP-Pd-PEPPSI-2**. The synthetic procedure was similar to that of KAP-Pd-PEPPSI-1. Briefly, a mixture of 1,3-Bis(2,6-diisopropyl-4-(chloromethyl) phenyl)-4,5-dihydro-1H-imidazol-3-iumchloride (524 mg, 1 mmol), anhydrous FeCl₃ (1.95 g, 12 mmol), and triphenylmethane (163 mg, 0.67 mmol) in anhydrous

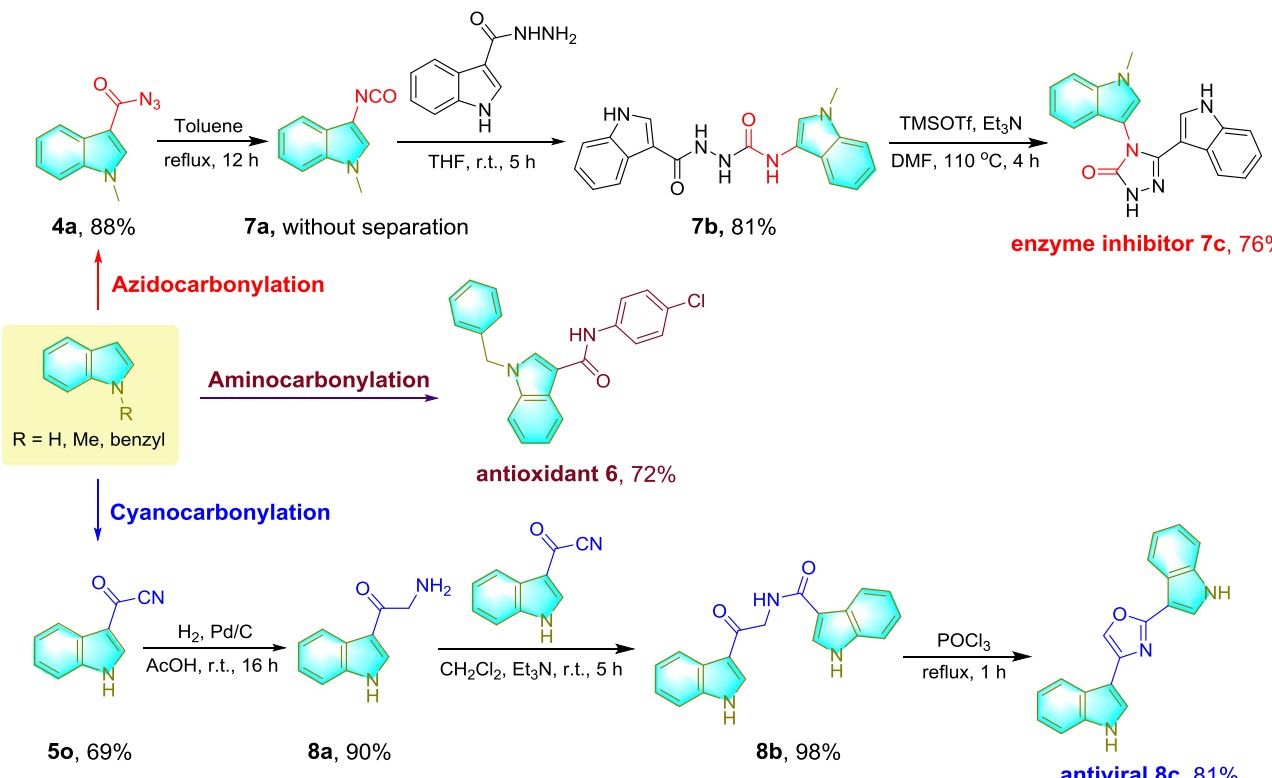

**Fig. 6 Synthetic applications.** The synthesis of representative bioactive molecules via KAP-Pd-PEPPSI catalyzed carbonylations of indoles.

1,2-dicholoroethane was refluxed at 80 °C under N₂ for 24 h to afford the KAP-2 in 72% yield, then the coordination of PdCl₂ with the NHC sites in KAP-2 delivered the KAP-Pd-PEPPSI-2 in 91% yield.

**Synthesis of KAP-Pd-PEPPSI-3.** The synthetic procedure was similar to that of KAP-Pd-PEPPSI-1. Briefly, a mixture of 1,3-Bis(2,6-diisopropyl-4-(chloromethyl) phenyl)-4,5-dihydro-1H-imidazol-3-iumchloride (524 mg, 1 mmol), anhydrous FeCl₃ (1.95 g, 12 mmol), and tetraphenylmethane (160 mg, 0.5 mmol) in anhydrous 1,2-dicholoroethane was refluxed at 80 °C under N₂ protection for 24 h to afford the KAP-3 in 82% yield. Subsequently, the complexation of PdCl₂ with the NHC sites in KAP-2 produced the KAP-Pd-PEPPSI-3 in 91% yield.

**Synthesis of KAP-Pd-PPh₃.** Triphenylphosphine hydrobromide (229.9 mg, 0.67 mmol), anhydrous FeCl₃ (1.95 g, 12 mmol), and 1,4-Bis(chloromethyl)benzene (175.1 mg, 1 mmol) were mixed together in anhydrous 1,2-dicholoroethane. The mixture was refluxed at 80 °C under N₂ for 24 h, and the resultant precipitates were collected by filtration and washed continually with methanol until the filtrate was clear and then treated by Soxhlet extraction in methanol for 24 h. The obtained reddish brown powder was redispersed in 10 mL MeOH and KOH (56 mg, 1 mmol) in 5 mL MeOH were slowly added. After stirring this mixture at room temperature for 2 h, the solid was collected by filtration and dried under vacuum to afford the crosslinked polymer with built-in phosphine (KAP-PPh₃, 89% yield). Then, a mixture of KAP-PPh₃ (384 mg) and bis(acetonitrile)palladium(II) chloride (36 mg, 0.14 mmol) in 10 mL CH₂Cl₂ was stirred at room temperature for 10 h, after which the precipitates were collected by filtration and washed with MeCN and CH₂Cl₂. Further drying under vacuum at 60 °C for 24 h gave the KAP-Pd-PPh₃ catalyst in 92% yield.

**Synthesis of POP-Pd-PEPPSI.** 1,3-bis(4-iodo-2,6-diisopropylphenyl)-4,5-dihy-dro-1H-imidazol-3-iumchloride (488 mg, 0.72 mmol) and 1,3,5-triethynylbenzene (72 mg, 0.48 mmol) were dissolved in a mixture of toluene (150 mL) and diiso-propylamine (6 mL). After adding bis(triphenylphosphine)palladium(II) chloride (30 mg, 0.042 mmol) and CuI (30 mg, 0.159 mmol), the reaction mixture was stirred at 90 °C for 24 h, after which the solid was retrieved by filtration and washed with excess acetone and methanol. The resultant polymer was dried under vacuum for 24 h at 60 °C (66% yield). Then, the complexation of PdCl₂ with the NHC sites in above polymer delivered the POP-Pd-PEPPSI in 96% yield.

**Synthesis of Pd@PC.** A mixture of sucrose (12.5 g), phytic acid solution (5 g) and silica colloid (20 g) in 50 g ultrapure water was stirred at room temperature for 12 h to give a homogeneous slurry. After being dried at 100 °C for 8 h, the obtained solid

was further heated at 160 °C for 8 h. The obtained black lumps were grounded into fine powder and then transferred into a tube furnace for carbonization under nitrogen. The temperature was ramped linearly from r.t. to 600 °C at a heating rate of 2 °C min⁻¹, then raised to 900 °C at a heating rate of 5 °C min⁻¹ and maintained at 900 °C for another 3 h. Upon cooling to room temperature, the obtained solid (denoted as carbon@silica) was treated with hydrofluoric acid (10 wt%) to remove the silica nanoparticles and the resultant powder was washed thoroughly with ultrapure water. After being dried under vacuum at 100 °C for 12 h, P-doped mesoporous carbon (PC) was obtained. Then, 1.0 g of the PC was added into an impregnation solution of PdCl₂ (34 mg, 0.28 mmol) in HCl solution (20 mL, 0.5 wt %), and the mixture was stirred at 80 °C for 12 h. The resultant solid was collected and dried at 100 °C for 24 h, and further reduced under a flow of H₂ at 200 °C for 2 h (with a heating rate of 2 °C min⁻¹) to afford the Pd@PC.

**Synthesis of Pd@NC.** A mixture of sucrose (12.5 g), cyanamide (5 g), and silica colloid (20 g) in 50 g ultrapure water was stirred at room temperature for 12 h to give a homogeneous slurry. The procedures of drying, carbonization, template removing, impregnation, and reduction were similar to that of PC.

**Synthesis of Pd@SiO₂.** PdCl₂ (34 mg, 0.5 wt%), then 1.0 g of the large pore silica gel (380-550 m²g⁻¹) were added into this impregnation solution. After stirring at 80 °C for 12 h, the resulting solids were dried at 100 °C for 24 h and calcined at 400 °C in air for 4 h. The resultant catalysts were later reduced under a flow of H₂ at 200 °C for 2 h (with a heating rate of 2 °C min⁻¹).

**Materials characterization.** ¹³C CP/MAS NMR measurements were performed using a Bruker 400 M MAS system with Adamantane as internal reference for C. Liquid NMR spectra were recorded using a Bruker Avance TM spectrometer operating at 400 MHz for ¹H and 100 MHz for ¹³C. XPS were carried out on a ESCALAB 250Xi X-ray photoelectron spectroscopy. BET surface area was per-formed on Micromeritics ASAP 2020 Physisorption analyzer. Before analysis, the catalysts were degassed at 120 °C for 6 h. The morphologies of catalyst were observed by SEM (Tecnai G2 F20 S-TWIN). STEM patterns were obtained using a FEI F20 with an acceleration voltage of 200 kV. FT-IR were performed on a Thermo Fisher Scientific Nicolet 6700 spectrometer. The contents of Pd in catalysts and solution were measured with Thermo Fisher ICAP RQ instrument. TEM images of catalysts were obtained with FEI Tecnai G² F20 S-Twin electron microscope at an acceleration voltage of 200 kV.

**Calculation of swelling degree**. Swelling degrees of KAP-Pd-PEPPSI in DMF are calculated based on equilibrium swelling method.

$$Q = \left(\frac{W1}{\rho 1} + \frac{W2}{\rho 2}\right) / \frac{W1}{\rho 1} \qquad (1)$$

Here, $W1$ and $W2$ are the polymer and the adsorbed solvent weights after swelling in DMF at 80 °C for 72 h, and $\rho$ represents the densities of polymer ($\rho 1$) and DMF ($\rho 2$) at room temperature. The swelling degree of KAP-Pd-PEPPSI-1, KAP-Pd-PEPPSI-2 and KAP-Pd-PEPPSI-3 calculated here are 239%, 171%, and 127%, respectively.

**General procedure for fluorocarbonylation of indole**. A mixture of **1a** (0.25 mmol), $I_2$ (0.5 mmol), DABCO (0.5 mmol) in 1 mL DMF was added into a 100 mL stainless steel autoclave with a magnetic bar. The mixture was stirred at room temperature for 60 min under $N_2$ protection. Then CsF (1 mmol), catalyst (5 mol%), DMF (1 mL) were added successively. Then the closed autoclave was vacuumed and purged with CO three times before it was finally pressurized with 2.0 MPa CO. Then the reactor was immersed in an oil bath preheated at 80 °C. After reaction for 24 h, the mixture was cooled down before excess CO was discharged. Finally, the resultant reaction mixture was analyzed by HPLC (Agilent 1260 Infinity with an Agilent ZORBAX C18 column, methanol/$H_2O$ = 90/10, 0.8 mL/min, $\lambda$ = 268 nm) using biphenyl as an inner standard.

For the recycling study in Fig. 2, the fluorocarbonylation of indole was performed with the same reaction condition as described above except using recycled catalyst. After each completion of the reaction, the solid catalyst was collected and washed with 20 mL DMF, 20 mL water and 20 mL methanol to remove the organic and inorganic residues. Finally, the resultant catalyst was dried under vacuum at room temperature for 24 h, weighted, and reused in the next run.

For the recycling study in Fig. 3, the amount of **1a** and $I_2$ were increased to 0.5 mmol and 0.6 mmol while keeping other parameters unchanged. After reaction for 16 h, the solid catalyst was collected and washed with DMF, water and methanol to remove the organic and inorganic residues. Finally, the collected catalyst was dried under vacuum at room temperature for 24 h, weighted, and reused in the next run.

**General procedure for aminocarbonylation of indoles**. A mixture of indole (0.25 mmol), $I_2$ (0.5 mmol), DABCO (0.5 mmol) in 1 mL DMF was added into a 100 mL stainless steel autoclave with a magnetic bar. The mixture was stirred at room temperature for 60 min. Then CsF (1 mmol), catalyst (5 mol%), amine (1.25 mmol), DMF (1 mL) were added successively. Then the closed autoclave was vacuumed and purged with CO three times before it was finally pressurized with 2.0 MPa CO. Then the reactor was immersed in an oil bath preheated at 80 °C. After 24 h, the mixture was cooled down before excess CO was discharged. Finally, the resultant reaction mixture was purified by chromatography using silica gel (petroleum ether and ethyl acetate) to afford the corresponding product. The obtained products were qualitatively analyzed by liquid NMR (Zhongke-Niujin 400) and high resolution mass spectra (HRMS, Thermo Fisher Scientific LTQ FT Ultra with DART positive mode).

**General procedure for azidocarbonylation of indoles**. A mixture of indole (0.25 mmol), $I_2$ (0.5 mmol), DABCO (0.5 mmol) in 1 mL DMF was added into a 100 mL stainless steel autoclave with a magnetic bar. The mixture was stirred at room temperature for 60 min. Then CsF (1 mmol), catalyst (5 mol%), DMF (1 mL) were added successively. Then the closed autoclave was vacuumed and purged with CO three times before it was finally pressurized with 2.0 MPa CO. Then the reactor was immersed in an oil bath preheated at 80 °C. After 24 h, the mixture was cooled down before excess CO was discharged. Then $NaN_3$ (1 mmol) was added directly into the resulted mixture with no need for purification. The mixture was stirred at room temperature for 7 h. Finally, the resultant reaction mixture was purified by chromatography using silica gel (petroleum ether and ethyl acetate) to afford the corresponding product. The obtained products were qualitatively analyzed by liquid NMR and HRMS.

**General procedure for cyanocarbonylation of indoles**. A mixture of indole (0.25 mmol), $I_2$ (0.5 mmol), DABCO (0.5 mmol) in 1 mL DMF was added into a 100 mL stainless steel autoclave with a magnetic bar. The mixture was stirred at room temperature for 60 min. Then CsF (1 mmol), catalyst (5 mol%), DMF (1 mL) were added successively. Then the closed autoclave was vacuumed and purged with CO three times before it was finally pressurized with 2.0 MPa CO. Then the reactor was immersed in an oil bath preheated at 80 °C. After 24 h, the mixture was cooled down before excess CO was discharged. Then TMSCN (1 mmol) and CsF (1 mmol) were added directly into the resulted mixture with no need for purification. The mixture was stirred at room temperature for 7 h. Finally, the resultant reaction mixture was purified by chromatography using silica gel (petroleum ether and ethyl acetate) to afford the corresponding product. The obtained products were qualitatively analyzed by liquid NMR and gas chromatography in combination with mass spectrometry (GC-MS, Agilent 5975 C/7890 A equipped with a HP-5MS column).

**General procedure for synthesis of antioxidant 6**. 1-benzyl-N-(4-chlorophenyl)-1H-indole-3-carboxamide **6** was synthesized via aminocarbonylation of N-benzylindole with 4-chloroaniline using a one-pot two-step procedure. Following the fluorocarbonylation of indole, 4-chloroaniline added into the reaction mixture, then the mixture was stirred at 60 °C for 12 h. The crude product was extracted with ethyl acetate and washed with deionized water, then purified by recrystallization from ethanol. The obtained products were qualitatively analyzed by liquid NMR and HRMS.

**General procedure for synthesis of enzyme inhibitor 7c**. 4,5-bisindolylpyrazolone **7c** was synthesized form 1-methyl-indole-3-carbonyl azide via azidocarbonylation of 1-methylindole. Nucleophilic addition of indole-3-carbohydrazide to 3-isocyanato-N-methylindole **7a** that was produced by one-pot azidocarbonylation of N-methylindole and then refluxing in toluene, gave bisindole carbohydrazide **7b**, following which cyclization of **7b** afforded **7c**.

**General procedure for synthesis of antiviral 8c**. Nortopsentin analogue **8c** was synthesized form indole-3-carbonyl cyanide via cyanocarbonylation of indole. Nucleophilic substitution of **5o** with oxotryptamine **8a**, which was synthesized from hydrogenation of another molecule of **5o**, produced indole-3-carboxamide **8b**, and its subsequent cyclization delivered **8c**.

**Reporting summary**. Further information on research design is available in the Nature Research Reporting Summary linked to this article.

## Data availability
The data supporting the findings of this study are available within the article and its Supplementary Information files. Source data are provided with this paper.

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

## Acknowledgements

This work was supported by National Key R&D Program of China (2018YFB1501600), Natural Science Foundation of China (21603246, 21972151), Light of West China of Chinese Academy of Sciences (CAS), DNL Cooperation Fund, CAS (DNL180303), and Key Research Program of Frontier Sciences of CAS (QYZDJSSW-SLH051).

## Author contributions

C.Y. and Q.X. conducted the experiments, analyzed the results, and wrote the manuscript. F.L. designed the research, supervised the project, and edited the manuscript. P.S. and H.L. participated in writing the manuscript. Z.Z. performed the SEM experiments and data analysis.

## Competing interests

The authors declare no competing interests.
