## [Peer Review File · Nature Communications]

REVIEWER COMMENTS

Reviewer #1 (Remarks to the Author):

The present article reports the results of an interesting study for Pd catalyst design using hyper-crosslinked polymer as a support. Excellent stabilization of the metal species by the applied polymeric support was demonstrated by the experiment (Fig. 4a and text). The catalyst was recycled many times and successfully re-used.

Stabilization of cocktail type system within the polymeric matrix is a new conceptual advance of the present study. The idea is highly useful and can potentially lead to the development of new type of heterogenized ligands for a broad range of reactions. The utility of the developed catalytic system was demonstrated with a good substrates scope. The study is well done and supplemented in details with electron microscopy characterization.

In overall, this is a high impact study and important methodology, which deserves publication after revision and clarification of the following points.

- 1) While the present study shows a good advance in the polymeric support design, extra claims should be avoided. Pd dynamics in the Pd/NHC systems is known and should be properly mentioned (see, for example, Chem. Sci., 2020,11, 6957-6977; etc.).
- 2) Fig. 5c shows a key issue dealing with the catalyst stabilization. A mismatch between the size of Pd atoms and light atoms (C,N) appears in the figure. What is the size of a polymeric cage for Pd atoms and clusters to fit inside? I suggest to use space filling model to see the actual size of Pd atoms and correct proportions. Perhaps, this figure should be redrawn to avoid ambiguous representation.
- 3) On Fig. 5c polymeric molecular fence is shown as a closed cage. While molecular structure of KAP-Pd-PEPPSI-3 is not a cage (Fig. 2a). Cross-linking influence is not evident and stays hidden, despite mentioned in the title. This should be commented or corrected.
- 4) Different cross-linking degree is mentioned throughout the manuscript. But how the cross-linking degree was controlled? What is the value of cross-linking degree? What are the estimates of the polymer weight?
- 5) Pd amount of 5 mol% is acceptable, although somewhat large for practical synthetic procedures. Optimizations of catalyst amount with 3, 1, 0.5 mol% of the catalyst should be performed and discussed.

Reviewer #2 (Remarks to the Author):

Li and co-workers report on an interesting heterogeneous catalyst where palladium is trapped inside

cross-linked polymers that are functionalized by heterocyclic carbene (NHC). The NHC-functionalized polymers lead to a useful stabilization of the palladium, and the best catalyst can be recycled many times without loss of activity in the reactions tested. The catalyst is successfully tested in a number of carbonylation reactions as well as in the Suzuki-Miyara coupling. The catalysts developed and tested are well characterized, new organic compounds are well characterized, and the results obtained are interesting and important. The work is of high quality. I recommend publication in Nature Communication after some minor revision.

1. It is not clear to the reader until p. 4 that 3-iodo-indole is first formed which is used in an oxidative addition. Initially, as a reader one thinks that there is a C-H activation by the catalyst (in fact the paper is written like this in the beginning). This should be clarified early in the manuscript
2. p. 4, right column, last paragraph: the end of the 1st sentence has to be rewritten
3. Table 4, Compound 5j does not fit into the Table since there is not nitrogen in the 7-position in the general top reaction
4. In general the manuscript is well written, but needs some minor linguistic improvement. I leave that to the Editorial Office. In several instances they have used "stabilizing mechanism". It is not clear what they mean by that. Please rewrite.

Reviewer #3 (Remarks to the Author):

The manuscript described leaching-deposition catalysis of supported Pd with N-heterocyclic carbene-functionalized hypercrosslinked polymers and the application to carbonylation and the Suzuki coupling. The reviewer cannot recommend its publication in Nat. Commun. by the following reasons:

1. The authors performed the carbonylation of indoles. As shown in Fig.4d, the optimal yield of the product was only ~50% using the best catalyst KAPs-Pd-PEPPSI-3, which is not efficient. To say worse, 2.5-5 mol% of Pd was used, whose catalytic efficiency is definitely low. It is not sufficient in term of synthetic utility.
2. The authors showed the carbonylation at 3-position of indoles that does not seem important. The reviewer cannot find any significance of the utility of the reaction. The authors showed the Suzuki coupling. However, only the reaction of 4-iodobenzene and phenylboronic acid was carried out. The coupling of aryl iodides is already well-developed, and the reviewer cannot find any merits in the result. These results lack superiority to be published in Nat. Commun.
3. The authors showed Pd leaching in Table 1. Leaching level by KAP-Pd-PEEPSI is still too high (0.2%). Recent heterogeneous catalytic systems care about ppm level of leaching. The reviewer cannot find any importance in the results.

Other items of the reviewer's concern are shown below:

1. The resolution of the XPS spectra is too low for valence analysis. Nevertheless, the authors have forced the assignment. They are too arbitrary and cannot be accepted as data.

2. Although it is important to confirm the state of palladium when the catalyst is supported on the support in this system, the surface of the support was only observed by low resolution STEM, which does not reveal the state of palladium aggregation. High-resolution TEM observation is required.

Responses to Reviewers' comments

Reviewer #1

Comment. *The present article reports the results of an interesting study for Pd catalyst design using hyper-crosslinked polymer as a support. Excellent stabilization of the metal species by the applied polymeric support was demonstrated by the experiment (Fig. 4a and text). The catalyst was recycled many times and successfully re-used.*

Stabilization of cocktail type system within the polymeric matrix is a new conceptual advance of the present study. The idea is highly useful and can potentially lead to the development of new type of heterogenized ligands for a broad range of reactions. The utility of the developed catalytic system was demonstrated with a good substrates scope. The study is well done and supplemented in details with electron microscopy characterization.

In overall, this is a high impact study and important methodology, which deserves publication after revision and clarification of the following points.

Response. We appreciate reviewer very much for the positive comment and encouragement.

Comment 1. *While the present study shows a good advance in the polymeric support design, extra claims should be avoided. Pd dynamics in the Pd/NHC systems is known and should be properly mentioned (see, for example, Chem. Sci., 2020, 11, 6957-6977; etc.).*

Response 1. Thank the reviewer very much for this important reminder. Following this suggestion, we have properly discussed the Pd dynamics of the Pd/NHC systems in our revised manuscript by reference to this important review (*Chem. Sci.* 2020, 11, 6957-6977) as well as some closely related articles (*Catal. Sci. Technol.* 2020, 10, 1228-1247; *Inorg. Chem.* 2019, 58, 12218-12227)¹⁻³. It has been found that the molecular Pd/NHC complexes may undergo Pd-C_{NHC} bond cleavage during catalysis to produce either NHC coordinated or ligandless active Pd species, leading to dynamic catalytic processes. For KAP-Pd-PEPPSI catalyzed heterogeneous carbonylations of indole, this Pd dynamics was also involved in the reaction process. Our investigations have demonstrated that this reaction system underwent a catalyst activation to produce NHC-coordinated Pd(0) clusters followed by an oxidative addition-reductive elimination to afford the product, presenting a Pd dissolution and redeposition behavior. Based on this, we revised our manuscript in page 6 as follows: "To get more insights into this KAPs-supported Pd catalyzed carbonylative C-C bond formation reaction under solid-liquid condition, the time-course analyses of Pd evolution and distribution as well as corresponding reaction activity in fluorocarbonylation of indole were carried out using typical Pd@C and the optimized KAPPd-PEPPSI-3 as the catalyst, respectively. As revealed in recent studies that

the molecular Pd/NHC complexes may undergo a dynamic process during the catalysis by Pd-C_{NHC} bond cleavage to produce either NHC coordinated or ligandless active Pd species⁶⁰⁻⁶². Here, the KAP-Pd-PEPPSI-3 experienced a long activation time to reduce the polymeric Pd(II)-NHC precatalyst to the active NHC-coordinated Pd(0) clusters...”

References

1. Chernyshev, V. M., Denisova, E. A., Eremin, D. B., Ananikov, V. P. The key role of R-NHC coupling (R = C, H, heteroatom) and M-NHC bond cleavage in the evolution of M/NHC complexes and formation of catalytically active species. *Chem. Sci.* **11**, 6957-6977 (2020).
2. Khazipov, O. V. *et al.* Preventing Pd-NHC bond cleavage and switching from nano-scale to molecular catalytic systems: amines and temperature as catalyst activators. *Catal. Sci. Technol.* **10**, 1228-1247 (2020).
3. Denisova, E. A., Eremin, D. B., Gordeev, E. G., Tsedilin, A. M., Ananikov, V. P. Addressing Reversibility of R-NHC Coupling on Palladium: Is Nano-to-Molecular Transition Possible for the Pd/NHC System? *Inorg. Chem.* **58**, 12218-12227 (2019).

As suggested, we have carefully revised our manuscript in Page 1 as the following:

“To get more insights into this KAPs-supported Pd catalyzed carbonylative C-C bond formation reaction under solid-liquid condition, the time-course analyses of Pd evolution and distribution as well as corresponding reaction activity in fluorocarbonylation of indole were carried out using typical Pd@C and the optimized KAPPd-PEPPSI-3 as the catalyst, respectively. **As revealed in recent studies that the molecular Pd/NHC complexes may undergo a dynamic process during the catalysis by Pd-C_{NHC} bond cleavage to produce either NHC coordinated or ligandless active Pd species⁶⁰⁻⁶².** Here, the KAP-Pd-PEPPSI-3 experienced a long activation time to reduce the polymeric Pd(II)-NHC precatalyst to the active NHC-coordinated Pd(0) clusters, the formation of which could be testified by the complete prohibition of the reaction after adding mercury (mercury poisoning experiment), and **A** started to be generated with its yield rapidly increasing to 87% in the next six hours (blue curve with dots).”

60. Chernyshev, V. M., Denisova, E. A., Eremin, D. B., Ananikov, V. P. The key role of R-NHC coupling (R = C, H, heteroatom) and M-NHC bond cleavage in the evolution of M/NHC complexes and formation of catalytically active species. *Chem. Sci.* **11**, 6957-6977 (2020).

61. Khazipov, O. V. *et al.* Preventing Pd-NHC bond cleavage and switching from nano-scale to molecular catalytic systems: amines and temperature as catalyst activators. *Catal. Sci. Technol.* **10**, 1228-1247 (2020).

62. Denisova, E. A., Eremin, D. B., Gordeev, E. G., Tsedilin, A. M., Ananikov, V. P. Addressing Reversibility of R-NHC Coupling on Palladium: Is Nano-to-Molecular Transition Possible for the Pd/NHC System? *Inorg. Chem.* **58**, 12218-12227 (2019).

Comment 2. *Fig. 5c shows a key issue dealing with the catalyst stabilization. A mismatch between the size of Pd atoms and light atoms (C, N) appears in the figure. What is the size of a polymeric cage for Pd atoms and clusters to fit inside? I suggest to use space filling model to see the actual*

size of Pd atoms and correct proportions. Perhaps, this figure should be redrawn to avoid ambiguous representation.

Response 2. We appreciate this helpful comment very much. As suggested, we used a space filling model to simulate the structure of KAP-Pd-PEPPSI in the model fluorocarbonylation of indole, and the original Figure 5c has been redrawn according to below investigations (Figure 5c). Structural simulation and optimization of this polymer were performed using molecular mechanics method (MM2). The size of the confined atoms was set as the inherit atom size in order to see the actual size of Pd atoms and ascertain the correct proportions of the involved components. Besides, the size of this polymeric cage was calculated with the method of matrix diagonalization using the optimized polymer structure¹. As shown in the following picture, the ideal polymeric cage in KAP-Pd-PEPPSI-3 featured a twisted adamantine structure. The side length of this cage was calculated to be 22.02 Å (Figure 1-1)², and the maximum diameter of its inscribed sphere was 38.14 Å (the diameter of Pd atom was 4.20 Å), indicating only small Pd nanoparticles or clusters can be formed inside the polymer. This result verified the confinement effect of the hypercrosslinked polymer on the supported Pd species.

References

1. Kopp, J. Efficient Numerical Diagonalization of Hermitian 3×3 Matrices. *Int. J. Mod. Phys. C* **19**, 523-548 (2008).
2. Mantina, M., Chamberlin, A. C., Valero, R., Cramer, C. J., Truhlar, D. G. Consistent van der Waals Radii for the Whole Main Group. *J. Phys. Chem. A* **113**, 5806-5812 (2009).

Figure 5c Proposed stabilizing effect of the polymeric support on active Pd species in the fluorocarbonylation of indole.

Figure 1-1 Size of the repeat molecular unit of the molecular cage.

Comment 3. On Fig. 5c polymeric molecular fence is shown as a closed cage. While molecular structure of KAP-Pd-PEPPSI-3 is not a cage (Fig. 2a).

Response 3. Thank you very much for this insightful comment. It has been reported that the use of tetraphenylmethane as the crosslinking group can produce an adamantane-like cage structure in the polymer¹⁻⁵. Herein, KAP-Pd-PEPPSI-3 was synthesized via the Friedel-Crafts alkylation polymerization of tetraphenylmethane with chloromethyl functionalized NHC precursor (Figure 1-2) and then coordination with palladium salt. Therefore, KAP-Pd-PEPPSI-3 featured a twisted adamantane-like cage structure as shown in the redrawn Figure 5c.

References

1. Evans, A. M. *et al.* Emissive Single-Crystalline Boroxine-Linked Colloidal Covalent Organic Frameworks. *J. Am. Chem. Soc.* **141**, 19728-19735 (2019).
2. Ma, T. *et al.* Observation of Interpenetration Isomerism in Covalent Organic Frameworks. *J. Am. Chem. Soc.* **140**, 6763-6766 (2018).
3. Li, H. *et al.* Three-Dimensional Covalent Organic Frameworks with Dual Linkages for Bifunctional Cascade Catalysis. *J. Am. Chem. Soc.* **138**, 14783-14788 (2016).
4. Baldwin, L. A., Crowe, J. W., Pyles, D. A., McGrier, P. L. Metalation of a Mesoporous Three-Dimensional Covalent Organic Framework. *J. Am. Chem. Soc.* **138**, 15134-15137 (2016).
5. Gao, Z. Z. *et al.* Water-Soluble 3D Covalent Organic Framework that Displays an Enhanced Enrichment Effect of Photosensitizers and Catalysts for the Reduction of Protons to H₂. *ACS Appl. Mater. Interfaces* **12**, 1404-1411 (2020).

Figure 1-2 Friedel-Crafts alkylation polymerization of tetraphenylmethane with chloromethyl functionalized NHC precursor.

The revised Fig. 5 has been added into page 7 of the revised manuscript as below:

Fig. 5 Reaction process investigation. (a) Kinetic investigation. (b) Hot filtration test. Under the optimized reaction conditions, the reaction mixture was quickly divided into two parts by centrifugation after reaction for 4 hours, and the upper clear liquid proceeded to react for another 20 h in the presence of CsF (c) Proposed stabilizing effect of the polymeric support on active Pd species in the fluorocarbonylation of indole.

Comment 4. Different cross-linking degree is mentioned throughout the manuscript. But how the cross-linking degree was controlled? What is the value of cross-linking degree? What are the estimates of the polymer weight?

Cross-linking influence is not evident and stays hidden, despite mentioned in the title. This should be commented or corrected.

Response 4. Thank you very much for this insightful comment. The crosslinking degree of different KAP-Pd-PEPPSI is controlled by changing nucleophilic monomers in the polymerization reactions¹. Friedel-Crafts alkylation polymerization of chloromethyl functionalized NHC precursor with three different nucleophiles (tetramethylbenzene, triphenylmethane and tetraphenylmethane) that varies the number of phenyl substituent produced the corresponding polymers with different crosslinking degrees. To give a visual description of the construction of the crosslinking structure, an ideal model of the polymerization reactions was provided in Figure 1-3. The reaction between NHC precursor and tetramethylbenzene generates a linear polymer with the lowest crosslinking degree (KAP-1). The reaction of NHC precursor with triphenylmethane produces a crosslinked polymer networks with a twisted plane structure (KAP-2). The polymerization of NHC precursor with tetraphenylmethane furnishes the hypercrosslinked polymer with an adamantane-like cage structure (KAP-3).

Figure 1-3 The synthesis of NHC-functionalized KAPs with different crosslinking degrees.

The most commonly used method for determining the degree of crosslinking is solvent extraction method which is based on extraction of the low molecular weight polymer from the initial polymer using a specific solvent and then calculation of the ratio of the final weight to the initial weight^{2,3}. Based on this, we tested the solubility of KAP-Pd-PEPPSI in common solvents such as ethanol, acetonitrile, xylene, dichloroethane and DMF. However, KAP-Pd-PEPPSI were totally insoluble in the above solvents, indicating a very large polymer weight. The complete insolubility of these polymers in solvent frustrated the determination of crosslinking degree using solvent extraction method and also excluded the possibility of testing the polymer weight by the common polymer weight determination method (e.g. gel permeation chromatography).

On the other hand, Froelich and Moeini et al. reported the use of equilibrium swelling method to calculate the crosslinking degree⁴⁻⁶. This method is based on the calculation of the equilibrium swelling ratio of a polymer in a specific solvent. For the same type of polymers, the crosslinking degree of a polymer depends inversely on its swelling degree⁷⁻⁹, and a higher swelling degree indicates a lower degree of crosslinking. Based on this, we employed the swelling degrees of KAP-Pd-PEPPSI to estimate their crosslinking degrees. Swelling degrees of KAP-Pd-PEPPSI in DMF are calculated based on equilibrium swelling method⁵:

$$Q = \left(\frac{W1}{\rho1} + \frac{W2}{\rho2} \right) / \frac{W1}{\rho1}$$

Here, $W1$ and $W2$ are the weights of the polymer and the adsorbed solvent after swelling in DMF at 80 °C for 72 h, and ρ represents the densities of polymer ($\rho1$) and DMF ($\rho2$) at room temperature. The swelling degree of KAP-Pd-PEPPSI-1, KAP-Pd-PEPPSI-2 and KAP-Pd-PEPPSI-3 calculated here are 239%, 171% and 127%, respectively. Therefore, KAP-Pd-PEPPSI-3 has a higher degree of crosslinking than KAP-Pd-PEPPSI-1 and KAP-Pd-PEPPSI-2.

To investigate the influence of crosslinking degree on the catalyst stability, we performed the recyclability test of different KAP-Pd-PEPPSI (Figure 4d). The results indicated that the stability of KAP-Pd-PEPPSI was highly dependent on its crosslinking degree, and higher crosslinking degrees led to better catalyst stability. Hypercrosslinked KAP-Pd-PEPPSI-3 provided an enhanced molecular

fence effect to the active Pd species which effectively suppressed the catalyst deactivation caused by Pd aggregation (Figure 4e).

References

1. Li, Z., Ding, X., Feng, Y., Feng, W., Han, B. H. Structural and Dimensional Transformations between Covalent Organic Frameworks via Linker Exchange. *Macromolecules* **52**, 1257-1265 (2019).
2. Wu, D. *et al.* In Situ Forming, Dual-Crosslink Network, Self-Healing Hydrogel Enabled by a Bioorthogonal Nopoldiol–Benzoxaborolate Click Reaction with a Wide pH Range. *Chem. Mater.* **31**, 4092-4102 (2019)
3. Kim, J. W., Choi, H. S. Surface crosslinking of high-density polyethylene beads in a modified plasma reactor. *J. Appl. Polym. Sci.* **83**, 2921-2929 (2010).
4. Zang, Y. H., Muller, R., Froelich, D. Determination of crosslinking density of polymer networks by mechanical data in simple extension and by swelling degree at equilibrium. *Polymer*, **30**, 2060-2062 (1989).
5. Moeini, H. R. Synthesis and properties of novel polyurethane-urea insulating coatings from hydroxyl-terminated prepolymers and blocked isocyanate curing agent. *J. Appl. Polym. Sci.* **112**, 3714-3720 (2009).
6. Hu, S., Chen, X., Torkelson, J. M. Biobased Reprocessable Polyhydroxyurethane Networks: Full Recovery of Crosslink Density with Three Concurrent Dynamic Chemistries. *ACS Sustain. Chem. Eng.* **7**, 10025-10034 (2019).
7. Gauthier, M. A. *et al.* Degree of Crosslinking and Mechanical Properties of Crosslinked Poly(vinyl alcohol) Beads for Use in Solid-Phase Organic Synthesis. *Polymer* **45**, 8201-8210 (2004).
8. Pacios, I. E., Molina, M. J., Gómez-Antón, M. R., Piérola, I. F. Correlation of swelling and crosslinking density with the composition of the reacting mixture employed in radical crosslinking copolymerization. *J. Appl. Polym. Sci.* **103**, 263-269 (2010).
9. Afinjuomo, F. *et al.* Synthesis and characterization of a novel inulin hydrogel crosslinked with pyromellitic dianhydride. *React. Funct. Polym.* **134**, 104-111 (2019).

The above mentioned Figure 1-1, 1-2 and 1-3 have been added into the revised Supplementary Information.

Comment 5. *Pd amount of 5 mol% is acceptable, although somewhat large for practical synthetic procedures. Optimizations of catalyst amount with 3, 1, 0.5 mol% of the catalyst should be performed and discussed.*

Response 5. Thank you very much for your kind advice. As suggested, fluorocarbonylations of indole with low catalyst loadings were performed using KAP-Pd-PEPPSI-2 as the catalyst, and the catalytic results are summarized in below Table 1-1. 3 mol% of KAP-Pd-PEPPSI-2 produced 88% yield of indole-3-carbonyl fluoride (**A**) (Entry 2), and 83% yield of **A** was obtained when lowering the catalyst amount to 2 mol% (Entry 3). Further reducing the catalyst loading to 1 mol% gave **A** in 54% yield (Entry 4), which could be promoted to 86% upon extending reaction time to 48 h (Entry 5).

Table 1-1. Optimizations of catalyst amount.^[a]

Entry	Catalyst amount (mol%)	Yield ^b (%)
1	5	89
2	3	88
3	2	83
4	1	54
5 ^c	1	86

^aReaction conditions: **1a** (0.25 mmol), CsF (1 mmol), I₂ (0.5 mmol), base (0.5 mmol), DMF (2 mL), 80 °C, 24 h, 2.0 MPa. ^bHPLC yield. ^c48 h.

Based on the above results, we have revised corresponding discussion in Page 4 as the following:

“Initial screening of reaction conditions was conducted with Pd-catalyzed fluorocarbonylation of 1-methyl-1H-indole (**1a**) using KAP-Pd-PEPPSI-2 as the precatalyst (Supplementary Table 2). After a series of primary investigations, the yield of indole-3-carbonyl fluoride (**A**) reached 89% when using CsF as the fluorine source and DABCO as the base (Supplementary Table 2, entry 5). **3 mol% of KAP-Pd-PEPPSI-2 produced 88% yield of indole-3-carbonyl fluoride (A), and 83% yield of A was obtained when lowering the catalyst amount to 2 mol% (Supplementary Table 2, Entry 12 and 13). Further reducing the catalyst loading to 1 mol% gave A in 54% yield, which could be promoted to 86% upon extending reaction time to 48 h (Supplementary Table 2, Entry 14 and 15).** Under the optimized reaction conditions as presented in Table 1...”

The data in above Table 1-1 have been added into the revised Supplementary Table 2 as following:

Supplementary Table 2. Optimization of fluorocarbonylation of indoles ^a

Entry	Solvent	[F]	Base	Yield (%)
1	DMF	CsF	Cs ₂ CO ₃	24
2	DMF	CsF	K ₂ CO ₃	12
3	DMF	CsF	K ₃ PO ₄	8
4	DMF	CsF	DBU	trace
5^b	DMF	CsF	DABCO	89(79)
6	DMF	KF	DABCO	74
7	DMF	NH ₄ F	DABCO	trace
8	DMF	NBu ₄ F	DABCO	32
9	Toluene	CsF	DABCO	trace
10	Dioxane	CsF	DABCO	trace
11 ^c	DMF	CsF	DABCO	trace

12 ^d	DMF	CsF	DABCO	88
13 ^e	DMF	CsF	DABCO	83
14 ^f	DMF	CsF	DABCO	54
15 ^g	DMF	CsF	DABCO	86

^a Conditions: 1a (0.25 mmol), CsF (1 mmol), catalyst (5 mol%), I₂ (0.5 mmol), base (0.5 mmol), DMF (2 mL), 80 °C, 24 h. ^b Isolated yield in bracket. ^c Mercury test. ^d 3 mol% catalyst. ^e 2 mol% catalyst. ^f 1 mol% catalyst. ^g 48 h.

Reviewer #2

Comment. *Li and co-workers report on an interesting heterogeneous catalyst where palladium is trapped inside cross-linked polymers that are functionalized by heterocyclic carbene (NHC). The NHC-functionalized polymers lead to a useful stabilization of the palladium, and the best catalyst can be recycled many times without loss of activity in the reactions tested. The catalyst is successfully tested in a number of carbonylation reactions as well as in the Suzuki-Miyara coupling. The catalysts developed and tested are well characterized, new organic compounds are well characterized, and the results obtained are interesting and important. The work is of high quality. I recommend publication in Nature Communication after some minor revision.*

Response. We appreciate reviewer very much for the positive comment and encouragement.

Comment 1. *It is not clear to the reader until p. 4 that 3-iodo-indole is first formed which is used in an oxidative addition. Initially, as a reader one thinks that there is a C-H activation by the catalyst (in fact the paper is written like this in the beginning). This should be clarified early in the manuscript.*

Response 1. Thank you very much for this important reminder which can help us improve the quality of this manuscript. As suggested, the formation of 3-iodo-indole before the carbonylation reaction has been clarified in the introduction part as following: In this work, we have developed a series of KAPs supported Pd complex precatalysts with different crosslinking structures (KAP-Pd-PEPPSI-x, x = 1, 2, 3, Fig. 2a), which showed remarkable activities in carbonylations of **the *in-situ* formed 3-iodo-indole via iodine-oxidation of indole C3-H bond** to corresponding valuable acyl fluoride, amides, aroyl azides and aroyl cyanides with significantly enhanced stability, surpassing other conventional supported Pd catalysts.

Subsequently, we have revised our manuscript in Page 2 as the following:

“The carbonylation reactions represent an important class of C-C bond formation reactions which can introduce carbonyl group into molecules and produce a variety of functional compounds³⁴. While ArX are widely used in Pd-catalyzed homogeneous carbonylation reactions, direct and heterogeneous carbonylation of unfunctionalized Ar-H has been rarely investigated due to the inertness of aryl C-H bond and poor catalyst recyclability under harsh and oxidative conditions³⁵. In this work, we have developed a series of KAPs supported Pd complex precatalysts with different crosslinking structures (KAP-Pd-PEPPSI-x, x = 1, 2, 3, Fig. 2a), which showed remarkable activities in carbonylations of **the *in-situ* formed 3-iodo-indole via iodine-oxidation of indole C3-H bond** to corresponding valuable acyl

fluoride, amides, aroyl azides and aroyl cyanides with significantly enhanced stability, surpassing other conventional supported Pd catalysts...

Comment 2, p. 4, right column, last paragraph: the end of the 1st sentence has to be rewritten.

Response 2. Thank you very much for this reminder. The first sentence in the last paragraph of page 4 “Then, we selected KAP-Pd-PPh₃, POP-Pd-PEPPSI and KAP-Pd-PEPPSI-2 which have similar skeletal structures to further study the recyclability of the supported Pd complex catalysts” has been changed into “Then, we selected KAP-Pd-PPh₃, POP-Pd-PEPPSI and KAP-Pd-PEPPSI-2 which have similar skeletal structures to study their recyclabilities in carbonylation reaction.”

Subsequently, we have revised our manuscript in Page 4 as the following:

“Then, we selected KAP-Pd-PPh₃, POP-Pd-PEPPSI and KAP-Pd-PEPPSI-2 which have similar skeletal structures to study their recyclabilities in carbonylation reaction. As shown in Fig. 3a, all these catalysts could be used at least four times without a significant decline in the activity...”

Comment 3. Table 4, Compound 5j does not fit into the Table since there is not nitrogen in the 7-position in the general top reaction

Response 3. We are highly appreciative of reviewer’s careful review and this kind reminder. As suggested, an X representing C or N was inserted into the molecular structure of indole, and corresponding reaction equations on the top of Table 2, Table 3 and Table 4 were all revised as followings:

Table 2. Aminocarbonylation of indoles and their analogues with aromatic amines^a.

Table 3. Azidocarbonylation of indoles and their analogues to synthesize indole-3-carbonyl azides^a.

Table 4. Cyanocarbonylation of indoles and their analogues to synthesize indole-3-carbonyl cyanides^a.

The above corrections have been added into the corresponding Table 2, Table 3 and Table 4 of the revised manuscript.

Comment 4. *In general the manuscript is well written, but needs some minor linguistic improvement. I leave that to the Editorial Office. In several instances they have used “stabilizing mechanism”. It is not clear what they mean by that. Please rewrite.*

Response 4. We appreciate very much the reviewer’s help and encouragement. The word “stabilizing mechanism” represents the stabilizing effects of the hypercrosslinked polymer on active Pd species. In order to better describe the influence of polymer structure on catalyst stability in terms of their confinement and electronic coordination and/or interaction, we have used “stabilizing effect of the hypercrosslinked polymer on active Pd species” and its abbreviation “stabilizing effect” to substitute the original word “stabilizing mechanism” throughout the manuscript.

Accordingly, we have revised our abstract in Page 1 as the followings:

“Catalyst deactivation caused by the aggregation of active metal species in the reaction process poses great challenges for practical applications of supported metal catalysts in solid-liquid catalysis. Herein, we developed a hypercrosslinked polymer integrated with N-heterocyclic carbene (NHC) as bifunctional support to stabilize palladium in heterogeneous C-C bond formations. This polymer supported palladium catalyst exhibits excellent stability in the one-pot fluorocarbonylation of indoles to four kinds of valuable indole-derived carbonyl compounds in cascade or sequential manner, as well as the representative Suzuki-Miyaura coupling reaction. **Investigations on stabilizing effect** disclose that this catalyst displays a molecular fence effect in which the coordination of NHC sites and confinement of polymer skeleton contribute together to stabilize the active palladium species in the reaction process. This work provides new insight into the development of supported metal catalysts with high stability and will also boost their efficient applications in advanced synthesis.”

Our manuscript in Page 2 as the followings:

“The carbonylation reactions represent an important class of C-C bond formation reactions which can introduce carbonyl group into molecules and produce a variety of functional compounds³⁴. While

ArX are widely used in Pd-catalyzed carbonylation reactions, direct and heterogeneous carbonylation of unfunctionalized Ar-H has been rarely investigated due to the inertness of aryl C-H bond and poor catalyst stability under harsh and oxidative conditions³⁵. In this work, we have developed a series of KAPs supported Pd complex precatalysts with different crosslinking structures (KAP-Pd-PEPPSI-x, x = 1, 2, 3, Fig. 2a), which showed remarkable activities in carbonylation of the *in-situ* formed 3-iodoindoles from indoles to corresponding valuable acyl fluoride, amides, aroyl azides and aroyl cyanides with significantly enhanced stability, surpassing other conventional supported Pd catalysts. **Investigations on the stabilizing effects revealed that** revealed that the KAP-Pd-PEPPSI displayed a molecular fence effect in which the coordination of the NHC sites and confinement of the hypercrosslinked polymer skeleton worked together to trap the active Pd species inside the swollen polymer in the reaction process, which effectively decreased the Pd content in solution and therefore prevented them from aggregation.”

Our manuscript in Page 7 as the followings:

“**Fig. 5 Reaction process investigation.** (a) Kinetic investigation. (b) Hot filtration test. Under the optimized reaction conditions, the reaction mixture was quickly divided into two parts by centrifugation after reaction for 4 hours, and the upper clear liquid proceeded to react for another 20 h in the presence of CsF (c) **Stabilizing effect of the hypercrosslinked polymer on active Pd species in KAP-Pd-PEPPSI catalyzed fluorocarbonylation of indole.**”

“**The stabilizing effect based on the above experimental results is proposed (Fig. 5c).** First, the built-in polymeric Pd(II) complex is *in-situ* reduced to NHC coordinated Pd(0) clusters, wherein the NHC group acts as an electron-donating ligand that increases the electron density of Pd(0), thereby stabilizing the Pd(0) clusters from aggregation⁶². Next, the oxidative addition of Pd(0) species into the C-I bond of 3-iodo-1-methylindole, generated by iodination of C3-H bond of indole...”

Our manuscript in Page 10 as the followings:

“In summary, we have developed a series of efficient and highly stable supported Pd catalysts for different C-C bond formations under solid-liquid catalysis. For the first time, the synthesis of acid halide by heterogeneous carbonylation was established via KAP-Pd-PEPPSI catalyzed fluorocarbonylation of indole with enhanced catalytic stability surpassing other conventional supported Pd catalysts. **Investigations on the stabilizing effect revealed that** the hypercrosslinked polymer skeleton of the catalyst worked like a molecular fence to effectively trap the dissolved Pd species inside the swollen polymer by the electronic NHC coordination and polymeric confinement in the reaction process...”

Reviewer #3

Comment. *The manuscript described leaching-deposition catalysis of supported Pd with N-heterocyclic carbene-functionalized hypercrosslinked polymers and the application to carbonylation and the Suzuki coupling. The reviewer cannot recommend its publication in Nat. Commun. by the following reasons:*

Response. We appreciate the reviewer's thoughtful comments, which have urged us to further improve the manuscript quality. We hope that the following responses will answer your questions and finally seek your approval.

Comment 1. *The authors performed the carbonylation of indoles. As shown in Fig.4d, the optimal yield of the product was only ~50% using the best catalyst KAPs-Pd-PEPPSI-3, which is not efficient.*

Response 1. Thank you very much for this important comment. Reuse of a heterogeneous catalyst in the probe reaction under optimized reaction conditions is the most common way to estimate the catalyst stability, and recent studies have demonstrated that reusability test under a relatively lower product yield gave more convincing result about the catalyst stability^{1,2}. Therefore, to fully test the catalyst stability, recyclability of the catalysts was performed in two reaction systems: one is conducted at high product yield under the optimal conditions (Figure 3a, ~90% yield); the other one was performed at a lower product yield under a controlled reaction condition (Figure 4d, ~50% yield. It is worth noting that control of the yield of **A** at ~50% by lowering the catalyst loading and shortening the reaction time was aimed to test catalyst stability more accurately, and it didn't represent the true catalytic activity). In both cases, KAP-Pd-PEPPSI showed good stability in the one-pot synthesis of valuable acid fluoride.

References

1. Shen, Y., Zheng, Q., Zhu, H., Tu, T. Hierarchical Porous Organometallic Polymers Fabricated by Direct Knitting: Recyclable Single-Site Catalysts with Enhanced Activity. *Adv. Mater.* **32**, 1905950 (2020).
2. Li, X, *et al.* Cooperative Multifunctional Catalysts for Nitrene Synthesis: Platinum Nanoclusters in Amine-Functionalized Metal-Organic Frameworks. *Angew. Chem. Int. Ed.* **56**, 16371-16375 (2017).

Comment 2. *To say worse, 2.5-5 mol% of Pd was used, whose catalytic efficiency is definitely low. It is not sufficient in term of synthetic utility.*

Response 2. We appreciate the review for this insightful comment. As suggested, the catalytic activity of KAP-Pd-PEPPSI-2 with lower catalyst loading was investigated and the results were summarized in below Table 1-1. 3 mol% of KAP-Pd-PEPPSI-2 produced 88% yield of indole-3-carbonyl fluoride (**A**) (Entry 2), and 83% yield of **A** was obtained when lowering the catalyst amount to 2 mol% (Entry 3). Further reducing the catalyst loading to 1 mol% gave **A** in 54% yield (Entry 4), which could be promoted to 86% upon extending reaction time to 48 h (Entry 5). Similarly, Andersen et al. firstly reported the synthesis of acid chlorides via carbonylation of aryl iodides with CO by using 1~10 mol% Pd(P^tBu)₂ as the catalyst¹, and Manabe et al. used 3 mol% Pd(OAc)₂/Xantphos to catalyze the carbonylation of aryl halides (Ar-Br, Ar-I) and aryl triflates (Ar-OTf) with N-formylsaccharin as a special CO resource to afford acid fluoride². These two reports possibly indicate that carbonylation of aryl halides to produce acid halides need a considerably high loading of Pd catalyst to achieve an ideal product yield, due to the significant challenge in the reductive elimination of (L)(CO)Pd(COAr)X (L= ligand, X= halide) intermediate to release the product¹. Moreover, the KAP-Pd-PEPPSI catalyst could also be efficiently used in the Suzuki coupling of aryl halides (including aryl chlorides) with the arylboronic acid with a low catalyst loading (0.4 mol%) (Supplementary Table 7). These positive results show the potential of KAP-Pd-PEPPSI in synthetic utility.

Table 3-1. Optimizations of catalyst amount. ^[a]		
		
Entry	Catalyst amount (mol%)	Yield ^b (%)
1	5	89
2	3	88
3	2	83
4	1	54
5 ^c	1	86

^aReaction conditions: **1a** (0.25 mmol), CsF (1 mmol), I₂ (0.5 mmol), base (0.5 mmol), DMF (2 mL), 80 °C, 24 h, 2.0 MPa. ^bHPLC yield. ^c48 h.

References

1. Quesnel, J. S., Arndtsen, B. A. A Palladium-Catalyzed Carbonylation Approach to Acid Chloride Synthesis. *J. Am. Chem. Soc.* **135**, 16841-16844 (2013).
2. Ueda, T., Konishi, H., Manabe, K. Palladium-Catalyzed Fluorocarbonylation Using N-Formylsaccharin as CO Source: General Access to Carboxylic Acid Derivatives. *Org. Lett.* **15**, 5370-5373 (2013).

Based on the above results, we have revised corresponding discussion in Page 4 as the following:

“Initial screening of reaction conditions was conducted with Pd-catalyzed fluorocarbonylation of 1-methyl-1H-indole (**1a**) using KAP-Pd-PEPPSI-2 as the precatalyst (Supplementary Table 2). After a

series of primary investigations, the yield of indole-3-carbonyl fluoride (**A**) reached 89% when using CsF as the fluorine source and DABCO as the base (Supplementary Table 2, entry 5). 3 mol% of KAP-Pd-PEPPSI-2 produced 88% yield of indole-3-carbonyl fluoride (**A**) (Supplementary Table 2, entry 12), and 83% yield of **A** was obtained when lowering the catalyst amount to 2 mol% (Supplementary Table 2, entry 13). Further reducing the catalyst loading to 1 mol% gave **A** in 54% yield, which could be promoted to 86% upon extending reaction time to 48 h (Supplementary Table 2, entry 14 and 15). Under the optimized reaction conditions as presented in Table 1, all Pd-based heterogeneous catalysts provided **A** in high yields (82-93%, entries 2-10), indicating for the first time the availability of synthesizing acid halides by heterogeneous carbonylation....”

The data in above Table 1-1 have been added into the revised Supplementary Table 2 as following:

Supplementary Table 2. Optimization of fluorocarbonylation of indoles ^a

Reaction scheme: 1a (indole) reacts with KAPs-Pd-PEPPSI-2, I₂, base, DMF, 80 °C, CO, and [F] to produce **A** (indole-3-carbonyl fluoride).

Entry	Solvent	[F]	Base	Yield (%)
1	DMF	CsF	Cs ₂ CO ₃	24
2	DMF	CsF	K ₂ CO ₃	12
3	DMF	CsF	K ₃ PO ₄	8
4	DMF	CsF	DBU	trace
5 ^b	DMF	CsF	DABCO	89(79)
6	DMF	KF	DABCO	74
7	DMF	NH ₄ F	DABCO	trace
8	DMF	NBu ₄ F	DABCO	32
9	Toluene	CsF	DABCO	trace
10	Dioxane	CsF	DABCO	trace
11 ^c	DMF	CsF	DABCO	trace
12 ^d	DMF	CsF	DABCO	88
13 ^e	DMF	CsF	DABCO	83
14 ^f	DMF	CsF	DABCO	54
15 ^g	DMF	CsF	DABCO	86

^a Conditions: 1a (0.25 mmol), CsF (1 mmol), catalyst (5 mol%), I₂ (0.5 mmol), base (0.5 mmol), DMF (2 mL), 80 °C, 24 h. ^b Isolated yield in bracket. ^c Mercury test. ^d 3 mol% catalyst. ^e 2 mol% catalyst. ^f 1 mol% catalyst. ^g 1 mol% catalyst, 48 h.

Comment 3. *The authors showed the carbonylation at 3-position of indoles that does not seem important. The reviewer cannot find any significance of the utility of the reaction.*

Response 3. Thank the reviewer very much for this thoughtful comment which inspires us to explore the potential application of this reaction in pharmaceutical synthesis. It can be seen from literatures that indole-3-carbonyl derivatives are found in numerous natural products, and they are also among

the most prolific moieties in modern pharmaceutical molecules¹⁻⁵. For example, N-substituted indole-3-carboxamides have significant antioxidant activity⁶, bis-indole alkaloids such as bisindolylpyrazolones⁷, coscinamides⁸, dragmacidins⁹ and nortopsentins¹⁰ show potent biological activities as antitumor, antiviral, and antiinflammatory agents. Note that the above useful compounds are synthesized from the corresponding indole-3-carboxamides, indole-3-carbonyl azide or indole-3-carbonyl cyanide, which are usually prepared via a complicated carboxylation-chlorination-nucleophile substitution procedure from indoles using hazardous organolithium and halogenating reagents. Here, KAP-Pd-PEPPSI catalyzed one-pot carbonylations of indoles provide direct and sustainable ways to access these important compounds.

To illustrate the useful application of KAP-Pd-PEPPSI catalyzed carbonylations of indole in synthetic community, we performed the synthesis of 1-benzyl-N-(4-chlorophenyl)-1H-indole-3-carboxamide **6**⁶, 4,5-bisindolylpyrazolone **7c**⁷ and nortopsentin analogue **8c**¹¹, which show potent biological activities as antitumor, antiviral, and antiinflammatory agents, using the developed aminocarbonylation, azidocarbonylation and cyanocarbonylation as the key synthetic tools. As shown in Figure 3-1, aminocarbonylation of N-benzylindole with 4-chloroaniline via a one-pot two-step procedure generated **6** with a satisfactory yield (72%). Nucleophilic addition of indole-3-carbohydrazide to 3-isocyanato-N-methylindole **7a** that was obtained by one-pot azidocarbonylation of N-methyl indole and then refluxing in toluene, gave bisindole carbohydrazide **7b** in an overall yield of 81%, further cyclization of **7b** afforded **7c** in 76% yield. As to the application of indole-3-carbonyl cyanide generated by cyanocarbonylation of indole, nucleophilic substitution of **5o** with oxotryptamine **8a**, which was synthesized from hydrogenation of another molecule of **5o**, produced 98% yield of indole-3-carboxamide **8b**, and its subsequent cyclization delivered **8c** in 81% yield. These three representative syntheses demonstrate the outstanding potential of present KAP-Pd-PEPPSI catalyzed one-pot carbonylation in pharmaceutical synthesis.

References

1. Rajniak, J., Barco, B., Clay, N. K., Sattely, E. S. A new cyanogenic metabolite in Arabidopsis required for inducible pathogen defence. *Nature* **525**, 376-379 (2015).
2. Shiri, M. Indoles in Multicomponent Processes (MCPs). *Chem. Rev.* **112**, 3508-3549 (2012).
3. Feng, X., Liu, D., Li, Z., Bian, J. Bioactive modulators targeting STING adaptor in cGAS-STING pathway. *Drug Disco. Today* **25**, 230-237 (2020).
4. Ford, B. M., Tai, S., Fantegrossi, W. E., Prather, P. L. Synthetic Pot: Not Your Grandfather's Marijuana. *Trends in Pharmacol. Sci.* **38**, 257-276 (2017).
5. Nankar, R. P., Doble, M. Non-peptidyl insulin mimetics as a potential antidiabetic agent. *Drug Disco. Today* **18**, 748-755 (2013).
6. Ölgün S., Kiliç Z., Ada A. O., Çoban T. Synthesis and evaluation of novel N-H and N-substituted indole-2- and 3-carboxamide derivatives as antioxidants agents. *J. Enzym. Inhib. Med. Ch.* **22**, 457-462 (2007).

- Hu, Y. *et al.* Synthesis and biological evaluation of novel 4,5-bisindolyl-1,2,4-triazol-3-ones as glycogen synthase kinase-3 β inhibitors and neuroprotective agents. *Pharmazie* **72**, 707-713 (2017).
- Ragini, K., Piggott, A. M., Karuso, P. Bisindole Alkaloids from a New Zealand Deep-Sea Marine Sponge *Lamellomorpha strongylata*. *Mar. Drugs* **17**, 683 (2019).
- Capon, R. J. *et al.* Dragmacidins: New Protein Phosphatase Inhibitors from a Southern Australian Deep-Water Marine Sponge, *Spongosorites* sp. *J. Nat. Prod.* **61**, 660-662 (1998).
- Miyake F. Y., Yakushijin K., Horne D. A. A concise synthesis of topsentin A and nortopsentins B and D. *Org. Lett.* **2**, 2121-2123 (2000).
- Guo, J, *et al.* Optimization, Structure–Activity Relationship, and Mode of Action of Nortopsentin Analogues Containing Thiazole and Oxazole Moieties. *J. Agric. Food Chem.* **67**, 10018-10031 (2019).

Figure 6. The synthesis of representative bioactive molecules via KAP-Pd-PEPPSI catalyzed carbonylation of indoles.

Subsequently, we have revised our manuscript in Page 9 as follows:

To illustrate the useful application of KAP-Pd-PEPPSI catalyzed carbonylations of indole in synthetic community, we performed the synthesis of 1-benzyl-N-(4-chlorophenyl)-1H-indole-3-carboxamide **6**⁷⁶, 4,5-bisindolylpyrazolone **7c**⁷⁷ and nortopsentin analogue **8c**⁷⁸, which show potent biological activities as antioxidant, glycogen synthase kinase inhibitor and antiviral, using the developed aminocarbonylation, azidocarbonylation and cyanocarbonylation as the key synthetic tools. As shown in Figure 6, aminocarbonylation of N-benzylindole with 4-chloroaniline via a one-pot two-step procedure generated **6** with a satisfactory yield (72%). Nucleophilic addition of indole-3-carbohydrazide to 3-isocyanato-N-methylindole **7a** that was obtained by one-pot azidocarbonylation of N-methyl indole and then refluxing in toluene, gave bisindole carbohydrazide **7b** in an overall yield

of 81%, further cyclization of **7b** afforded **7c** in 76% yield. As to the application of indole-3-carbonyl cyanide generated by cyanocarbonylation of indole, nucleophilic substitution of **5o** with oxotryptamine **8a**, which was synthesized from hydrogenation of another molecule of **5o**, produced 98% yield of indole-3-carboxamide **8b**, and its subsequent cyclization delivered **8c** in 81% yield. These three representative syntheses demonstrate the outstanding potential of present KAP-Pd-PEPPSI catalyzed one-pot carbonylation in pharmaceutical synthesis.

76. Ölgün S., Kiliç Z., Ada A. O., Çoban T. Synthesis and evaluation of novel N-H and N-substituted indole-2- and 3-carboxamide derivatives as antioxidants agents. *J. Enzym. Inhib. Med. Ch.* **22**, 457-462 (2007).
77. Hu, Y. et al. Synthesis and biological evaluation of novel 4,5-bisindolyl-1,2,4-triazol-3-ones as glycogen synthase kinase-3 β inhibitors and neuroprotective agents. *Pharmazie* **72**, 707-713 (2017).
78. Guo, J. et al. Optimization, Structure-Activity Relationship, and Mode of Action of Nortopsentin Analogues Containing Thiazole and Oxazole Moieties. *J. Agric. Food Chem.* **67**, 10018-10031 (2019).

The Figure 6 has been added into Page 9 of the revised manuscript.

The corresponding characterization data for the products were provided at the end of this letter and in the supplementary information.

Comment 4. *The authors showed the Suzuki coupling. However, only the reaction of 4-iodobenzene and phenylboronic acid was carried out. The coupling of aryl iodides is already well-developed, and the reviewer cannot find any merits in the result. These results lack superiority to be published in Nat. Commun.*

Response 4. Thanks for this constructive comment. As suggested, representative couplings of unactivated aryl chlorides with phenylboronic acid were carried out using KAP-Pd-PEPPSI-3 as the catalyst. As can be seen from Supplementary Table 7, chlorobenzene was effectively reacted with phenylboronic acid to produce biphenyl in nearly quantitative yield (Entry 2), and introducing an electron-donating substituent (-CH₃) onto the benzene ring had no influence on the product yield (Entry 3). Representative N-heterocycles such as 3-chloropyridine and 5-chloroindole also gave the desired phenyl-substituted heterocyclic products in excellent yields (Entry 4 and 5). Interestingly, benzyl chloride, which is considered to be one of the most challenging coupling partners^{1,2}, could couple with phenylboronic acid efficiently to produce diphenylmethane in 92% yield (Entry 6).

References

1. Li, B. et al. Highly Dispersed Pd Catalyst Locked in Knitting Aryl Network Polymers for Suzuki-Miyaura Coupling Reactions of Aryl Chlorides in Aqueous Media. *Adv. Mater.* **24**, 3390-3395 (2012).

2. Wang, X. *et al.* Spatially Isolated Palladium in Porous Organic Polymers by Direct Knitting for Versatile Organic Transformations. *J. Catal.* **355**, 101-109 (2017).

Supplementary Table 7. KAP-Pd-PEPPSI catalyzed Suzuki coupling reactions of Aryl halides with phenylboronic acid^a.

^aReaction conditions: catalyst 0.4 mol%, aryl halide 1.0 mmol, aryl boronic acid 1.5 mmol, K₃PO₄ 3.0 mmol, EtOH/H₂O 4 mL, 80 °C, 6 h. ^bGC yield. ^c60 °C, 2 h, the Pd content in the filtrate after the completion of reaction was determined to be 0.06 ppm by ICP-MS.

Subsequently, we have revised our manuscript in Page 9 as the followings:

Besides efficient and stable applications in above carbonylative C-C bond formations, the optimized KAP-Pd-PEPPSI-3 also displayed excellent recyclability with up to 21 cycles and >95% product yield in another representative Suzuki-Miyaura coupling reaction at low catalyst loading (0.4 mol%) (Supplementary Fig. 13). STEM images of KAP-Pd-PEPPSI-3 after the 21st cycle showed uniformly distributed nanoparticles without formation of large aggregates (Supplementary Fig. 14c and 14d). XPS results revealed the presence of a strong metal-ligand interaction between Pd nanoparticles and NHC groups (Supplementary Fig. 15). On the contrary, typical Pd@C displayed a significant decrease in activity after the 6th cycle and the formation of large particle aggregates (up to 500 nm in size) suggests the severe Pd aggregation during the reaction (Supplementary Fig. 14a and 14b). These results further confirm the super stability of hypercrosslinked KAP-Pd-PEPPSI-3 and highlight its

potential applications in other C-C bond formation reactions. Interestingly, 0.4 mol% of KAP-Pd-PEPPSI-3 could also effectively catalyze the cross-coupling of representative unactivated phenyl chlorides and N-heterocyclic chlorides as well as the challenging benzyl chloride with phenylboronic acid into the corresponding biphenyls in excellent yields (92-98%, Supplementary Table 7).

The above mentioned Supplementary Table 7 has been added into the revised supplementary information.

The corresponding characterization data for the products were provided at the end of this letter and in the supplementary information.

Comment 5. *The authors showed Pd leaching in Table 1. Leaching level by KAP-Pd-PEPSI is still too high (0.2%). Recent heterogeneous catalytic systems care about ppm level of leaching. The reviewer cannot find any importance in the results.*

Response 5. Thank you very much for this thoughtful comment. Catalyst deactivation caused by the aggregation of active metal species in the reaction process poses great challenges for practical applications of supported metal catalysts in solid-liquid catalysis¹⁻⁴. Our initial investigations have revealed that the supported Pd catalyzed fluorocarbonylation of indole involved a Pd dissolution and redeposition process. Although most of the dissolved Pd could be redeposited onto the support after the completion of the reaction, severe Pd aggregations were found over the control catalysts (Pd@C, Pd@NC, Pd@PC, Pd@SiO₂, KAP-Pd-PPh₃ and POP-Pd-PEPPSI) and their electronic/geometric properties have changed a lot (Supplementary Figure 8 and Figure 3b), which thus resulted in the decreases in catalytic activity (Figure 3a). In contrast, the optimal KAP-Pd-PEPPSI-3 could effectively suppress the Pd aggregation via its molecular fence effect enabled by the coordination of NHC site and confinement of polymer skeleton, and it also showed a relatively lower level of Pd leaching (0.2%) than other heterogeneous Pd catalysts (up to 2.2%). Such good stability was also verified by the recyclability test of KAP-Pd-PEPPSI-3 catalyzed Suzuki-Miyaura reaction (Supplementary Figure 13 and 14), wherein the leaching level of Pd in the filtrate after the completion of the reaction was determined to be only 0.06 ppm (Supplementary Table 7, Entry 1), indicating an extremely low level of Pd leaching. These results highlight the outstanding stability of KAP-Pd-PEPPSI over the conventional supported Pd catalysts in the C-C bond formation reactions.

References

1. Biffis, A., Centomo, P., Del Zotto, A., Zecca, M. Pd Metal Catalysts for Cross-Couplings and Related Reactions in the 21st Century: A Critical Review. *Chem. Rev.* **118**, 2249-2295 (2018).
2. MacQuarrie, S. *et al.* Visual Observation of Redistribution and Dissolution of Palladium during the Suzuki-Miyaura Reaction. *Angew. Chem. Int. Ed.* **47**, 3279-3282 (2008).

- Scheuermann, G. M., Rumi, L., Steurer, P., Bannwarth, W., Müllhaupt, R. Palladium Nanoparticles on Graphite Oxide and Its Functionalized Graphene Derivatives as Highly Active Catalysts for the Suzuki-Miyaura Coupling Reaction. *J. Am. Chem. Soc.* **131**, 8262-8270 (2009).
- Ángeles Úbeda, M. *et al.* Precatalyst or Dosing-Device? The $[\text{Pd}_2\{\mu\text{-(C}_6\text{H}_4\text{)PPh}_2\}_2\{\mu\text{-O}_2\text{C(C}_6\text{H}_5\text{)}\}_2]$ Complex Anchored on A Carboxypolystyrene Polymer as An Effective Supplier of Palladium Catalytically Active Nanoparticles for The Suzuki-Miyaura Reaction. *J. Catal.* **381**, 26-37 (2020).

Other items of the reviewer's concern

Comment 1. *The resolution of the XPS spectra is too low for valence analysis. Nevertheless, the authors have forced the assignment. They are too arbitrary and cannot be accepted as data.*

Response 1. Thank you very much for this important comment. According to this suggestion, we have replaced the original XPS spectra in this manuscript with high-quality ones (Figure 3b and Figure 4b). The specific adjustments are as follows: 1. The XPS characterizations of the low-resolution samples were carried out again on a ESCALAB 250Xi X-ray photoelectron spectroscopy with increasing the number of scans to obtain high-resolution data; 2. The assignments of the Pd 3d binding energy peaks were refined based on our control experiments and related literatures; 3. The resolutions of XPS spectra were set to 300 dpi and the sizes of the updated XPS spectra in the revised manuscript were increased.

Figure 3b. Pd XPS spectra of different catalysts after being used for five times.

Figure 4b. Pd XPS spectra of recycled KAP-2/PdCl₂ and KAP-4/PdCl₂.

Meanwhile, the detailed assignment of Pd 3d photoelectron peaks has also been carefully illustrated. All XPS peak assignments in this work are based on an acceptable standard reference database and literature reports¹⁻³. The binding energy scale was set by fixing the C 1s component at 284.6 eV. The data analysis was performed with XPS Peak Fitting program (XPS Peak Fit 4.1). The Pd 3d binding energy peaks were decomposed using the Gaussian and Lorentzian function product model (least squares fitting) after subtraction of a nonlinear Shirley baseline. The molar concentration ratios of different Pd components were calculated from peak areas (linear background subtraction) normalized on the basis of the acquisition parameters after background subtraction. Determination of the Pd components was based on our control experiments with reference to the relevant literatures as follows:

Previous reports have demonstrated that the catalytic metal species involved in the supported Pd catalyzed C-C bond formation reactions had “Cocktail” character, which usually consisted of complicated Pd species (molecular complexes, clusters and nanoparticles)^{4,5}. Herein for KAP-Pd-PEPPSI catalyzed fluorocarbonylation of indole, Pd(II) species, Pd(0) clusters and nanoparticles over the support were also found to be involved in this system. The *in-situ* activation of Pd(II) complex generated NHC-coordinated Pd(0) clusters and aggregation of the above clusters further produced ligandless Pd(0) nanoparticles or aggregates.

The formation of NHC-coordinated Pd(0) clusters were verified by the control experiments (Figure 4a, 4b and 4c in the manuscript). Here, we used two structurally similar KAP supports with or without incorporated NHC precursor (KAP-2 and KAP-4) to interact with the *in-situ* formed Pd species from

PdCl₂ in fluorocarbonylation of indole to produce NHC-coordinated Pd(0) clusters and non-NHC-coordinated Pd(0) aggregates. The large aggregates observed in the TEM image of the KAP-4 support (Figure 4c, without NHC sites) showed the formation of the polymer supported Pd(0) aggregates with peak binding energy values of 335.60 and 340.85 eV for Pd 3d_{5/2} and Pd 3d_{3/2} (Figure 3-1, KAP-4/PdCl₂). The Pd 3d binding energy peaks of the Pd species (homogeneously distributed Pd species without formation of large aggregates, Figure 4c) over the KAP-2 support (with NHC sites) showed a negative shift of 0.51 eV in comparison to that of the Pd(0) aggregates, indicating the formation of NHC coordinated Pd(0) clusters with binding energy values of 335.09 and 340.34 eV for Pd 3d_{5/2} and Pd 3d_{3/2} (Figure 3-1, KAP-2/PdCl₂).

For heterogeneous Pd catalyzed cross-coupling reactions of aryl halides with coupling partners, previous reports have revealed that the oxidation of the Pd atoms on the surface of Pd(0) clusters generated the Pd(II) covered clusters^{6,7}. Based on this, oxidation of the *in-situ* formed Pd(0) clusters in KAP-Pd-PEPPSI using the reaction mixture (without coupling partner CsF) of fluorocarbonylation reaction was performed to identify the above Pd(II) species. As shown in Figure 3-1, after being treated at 80 °C for 24 h, the XPS spectrum of the obtained catalyst showed the characteristic Pd 3d core level peaks at 336.50 (Pd 3d_{5/2}) and 341.75 (Pd 3d_{3/2}) eV, conforming the formation of Pd(II) species via oxidation of Pd clusters.

Based on the reported literatures and the above control investigations, the Pd 3d binding energy peaks of the tested catalysts in this paper are assigned as followings: the peaks at 335.09 (Pd 3d_{5/2}) and 340.34 (Pd 3d_{3/2}) are assigned to ligand coordinated Pd(0) clusters, the peaks at 335.60 (Pd 3d_{5/2}) and 340.85 (Pd 3d_{3/2}) belong to Pd(0) aggregates, the peaks at 336.97 (Pd 3d_{5/2}) and 342.22 (Pd 3d_{3/2}) are attributable to the oxidized Pd(II) species, the peaks at 337.60 (Pd 3d_{5/2}) and 342.85 (Pd 3d_{3/2}) can be assigned to the Pd(II) species.

References

1. NIST X-ray Photoelectron Spectroscopy (XPS) Database, <https://srdata.nist.gov/xps/>.
2. Venezia, A. M. X-ray photoelectron spectroscopy (XPS) for catalysts characterization. *Catal. Today* **77**, 359-370 (2003)
3. Andersson C, Larsson R. Active sites in polymer-bound palladium-phosphine coordination catalysts. Chemical and XPS investigations. *J. Catal.* **81**, 194-203 (1983).
4. Eremin, D. B., Ananikov, V. P. Understanding Active Species in Catalytic Transformations: From Molecular Catalysis to Nanoparticles, Leaching, “Cocktails” of Catalysts and Dynamic Systems. *Coord. Chem. Rev.* **346**, 2-19 (2017).
5. Polynski, M. V., Ananikov, V. P. Modeling Key Pathways Proposed for the Formation and Evolution of “Cocktail”-Type Systems in Pd-Catalyzed Reactions Involving ArX Reagents. *ACS Catal.* **9**, 3991-4005 (2019).
6. Yuan, N. *et al.* Probing the Evolution of Palladium Species in Pd@MOF Catalysts during The Heck Coupling Reaction: An Operando X-ray Absorption Spectroscopy Study. *J. Am. Chem. Soc.* **140**, 8206-8217 (2018).

7. Reimann, S. *et al.* Identification of the Active Species Generated from Supported Pd Catalysts in Heck Reactions: An in situ Quick Scanning EXAFS Investigation. *J. Am. Chem. Soc.* **133**, 3921-3930 (2011).

Figure 3-1. Control experiments to investigate the Pd components involving in the supported Pd catalyzed fluorocarbonylation of indole. Blue areas: NHC coordinated Pd(0) clusters (Prepared by coordination of NHC-functionalized KAP with *in-situ* formed Pd(0) species form PdCl₂). Red areas: polymer supported Pd(0) aggregates (Prepared by deposition of *in-situ* formed Pd(0) aggregates form PdCl₂ onto non-NHC-functionalized KAP). Yellow areas: oxidized Pd clusters (Prepared by oxidation of *in-situ* formed Pd(0) clusters in KAP-Pd-PEPPSI-2 using the reaction mixture of fluorocarbonylation reaction in the absence of CsF). Purple areas: Pd(II) complexes (Pd(II) complexes in the original KAP-Pd-PEPPSI).

The above mentioned Figure 3-1 has been added into the revised supplementary information.

Comment 2. *Although it is important to confirm the state of palladium when the catalyst is supported on the support in this system, the surface of the support was only observed by low resolution STEM, which does not reveal the state of palladium aggregation. High-resolution TEM observation is required.*

Response 2. Thank you very much for this insightful comment which can help us improve the quality

of this manuscript. As suggested, the high-resolution TEM characterizations of different catalysts in the reusability tests were performed on a FEI Tecnai G² F20 S-Twin electron microscope at an acceleration voltage of 200 kV, and the obtained TEM images were added into the revised manuscript and the supplementary information as the key materials to reveal the state of palladium aggregation in catalyst. Detailed analyses about the catalyst structure based on the obtained TEM images were as follows:

Figure 3-2. TEM images of KAP-Pd-PPh₃, POP-Pd-PEPPSI and KAP-Pd-PEPPSI-2 after being used for five times. a-c: TEM images of different polymeric catalysts. d-f: TEM images of the recycled KAP-Pd-PEPPSI-2 at larger magnifications. g: STEM image of the recycled KAP-Pd-PEPPSI-2. h: EDS mapping images for Pd in KAP-Pd-PEPPSI-2 after the fifth cycle.

For the recyclability test of different polymeric catalysts in page 5, the TEM image of KAP-Pd-PPh₃ after five repeated uses showed large Pd aggregates with size up to 100 nm on the catalyst surface, and small aggregates with sizes ranging from several nanometers to dozens of nanometers were also observed in the TEM image of POP-Pd-PEPPSI (Figure 3-2, a and b). Whereas, TEM images of KAP-Pd-PEPPSI-2 exhibited a homogeneously distributed Pd component in catalyst and no large nanoparticles or aggregates could be found (Figure 3-2, c-h). These TEM images clearly demonstrated the state of the supported Pd in polymeric catalysts during the recyclability test, and also displayed significant differences in the Pd distribution between different catalysts. The TEM analyses were matched well with our recyclability test results in Figure 3a.

Figure 3-3. TEM images of recycled KAP-4/PdCl₂ (a) and KAP-2/PdCl₂ (b-d).

For the control experiment in page 6, very large Pd aggregates with size up to hundreds of nanometers were clearly observed in the TEM image of the recycled KAP-4/PdCl₂ (Figure 3-3, a). This TEM image provided a convinced evidence for the severe Pd aggregation in the ligandless catalyst system. On the other hand, TEM image of the recycled KAP-2/PdCl₂ displayed a relatively good dispersion of the Pd component (Figure 3-3, b-d). Only a small amount of Pd aggregates with size less than twenty nanometers were found on the support surface and most of the Pd were homogeneously distributed throughout the support. The stark contrast between the TEM images of the

recycled KAP-4/PdCl₂ and KAP-2/PdCl₂ in the Pd distribution highlighted the stabilizing effect of NHC sites on the Pd species.

Figure 3-4. TEM images of different KAP-Pd-PEPPSI-x after the 6th cycle. KAP-Pd-PEPPSI-1 (a), KAP-Pd-PEPPSI-2 (b), KAP-Pd-PEPPSI-3 (c).

For the reusability test of different KAP-Pd-PEPPSI in page 6, the TEM images of different KAP-Pd-PEPPSI after the 6th cycle gave a good presentation about the influence of the crosslinking degree on the Pd aggregation. As can be seen from Figure 3-4, the TEM image of the recycled KAP-Pd-PEPPSI-1 showed large amount of irregular Pd aggregates on the catalyst surface (Figure 3-4, a). Small amount of Pd aggregates with size less than twenty nanometers were also observed in the recycled KAP-Pd-PEPPSI-2 (Figure 3-4, b). Whereas, no observable aggregates were detected on the STEM image of the spent KAP-Pd-PEPPSI-3 (Figure 3-4, c). These results were highly consistent with the recyclability test results in Figure 4d. Besides, the obvious distinctions in the Pd distribution between the TEM images of different KAP-Pd-PEPPSI clearly demonstrated the positive effect of the hypercrosslinked structure on the catalyst stability.

The above mentioned Figure 3-2 and 4-3 have been added into the revised supplementary information.

Subsequently, we have revised our manuscript in Page 5 and Page 6 as the followings:

Fig. 3 Recyclability of the selected catalyst. (a) Recyclability of the supported Pd complex precatalysts in fluorocarbonylation of indole. Reaction conditions: **1a** (0.25 mmol), CsF (1 mmol), catalyst (5 mol%), I₂ (0.5 mmol), base (0.5 mmol), DMF (2 mL), 80 °C, 24 h, 2.0 MPa. (b) Pd XPS spectra and (c) TEM images of different catalysts after being used for five times. (d) Structures of the catalysts used here.

Fig. 4 Stabilizing effect of KAP. (a) Control experiments to determine the stabilizing effect of NHC under the optimized reaction conditions. (b) Pd XPS spectra and (c) TEM images of recycled KAP-2/PdCl₂ and KAP-4/PdCl₂. (d) Control experiments to determine the stabilizing effect of catalyst structure. Reaction conditions: **1a** (0.25 mmol), CsF (1 mmol), Pd catalyst (2.5 mol%), I₂ (0.3 mmol), base (0.5 mmol), DMF (2 mL), 80 °C, 16 h, 2.0 MPa. (e) TEM images of different KAP-Pd-PEPPSI-x after the 6th cycle.

Purified by recrystallization from ethanol; white solid; 72% yield. ^1H NMR (400 MHz, CDCl_3) δ 8.06 (dd, $J = 6.5, 1.9$ Hz, 1H), 7.78 (s, 1H), 7.69-7.56 (m, 3H), 7.42-7.27 (m, 8H), 7.21-7.12 (m, 2H), 5.36 (s, 2H). ^{13}C NMR (100 MHz, DMSO) δ 163.3, 139.2, 137.8, 136.8, 132.5, 129.2, 128.9, 128.1, 127.6, 127.4, 123.0, 121.9, 121.7, 111.3, 110.5. HRMS (ESI) m/z : calculated for $\text{C}_{22}\text{H}_{18}\text{ClN}_2\text{O}$ $[\text{M} + \text{H}]^+$ 361.1108, found 361.1112.

Purified by column chromatography (1:2 PE/EA); white solid; 76% yield; $R_f = 0.50$ (EA). ^1H NMR (400 MHz, DMSO) δ 11.74 (s, 1H), 10.98 (d, $J = 1.9$ Hz, 1H), 8.14-7.95 (m, 1H), 7.52 (s, 1H), 7.38 (d, $J = 8.5$ Hz, 1H), 7.19 (dd, $J = 6.5, 1.8$ Hz, 1H), 7.10-7.03 (m, 2H), 7.03-6.94 (m, 2H), 6.87 (t, $J = 7.4$ Hz, 1H), 6.42 (d, $J = 2.8$ Hz, 1H), 3.68 (s, 3H). ^{13}C NMR (100 MHz, DMSO) δ 155.6, 144.6, 136.0, 128.9, 125.4, 125.2, 124.8, 122.9, 122.6, 121.6, 120.9, 120.6, 117.9, 112.3, 111.0, 108.9, 102.7. HRMS (ESI) m/z : calculated for $\text{C}_{19}\text{H}_{16}\text{N}_5\text{O}$ $[\text{M} + \text{H}]^+$ 330.1355, found 330.1355.

Purified by column chromatography (2:1 PE/EA); yellow solid; 81% yield; $R_f = 0.58$ (3:1 PE/EA). ^1H NMR (400 MHz, DMSO) δ 11.77 (s, 1H), 11.59 (s, 1H), 8.29 (d, $J = 6.6$ Hz, 1H), 8.14 (d, $J = 2.2$ Hz, 1H), 7.97 (d, $J = 7.5$ Hz, 1H), 7.90 (d, $J = 1.9$ Hz, 1H), 7.51 (d, $J = 10.8$ Hz, 3H), 7.35-7.10 (m, 4H). ^{13}C NMR (100 MHz, DMSO) δ 157.3, 145.9, 136.9, 123.4, 121.1, 120.5, 120.1, 112.6, 104.6. HRMS (ESI) m/z : calculated for $\text{C}_{19}\text{H}_{13}\text{N}_3\text{ONa}$ $[\text{M} + \text{Na}]^+$ 322.0956, found 322.0961.

Purified by column chromatography (PE); white solid; > 98% yield; ^1H NMR (400 MHz, CDCl_3) δ 7.59 (dd, $J = 7.0, 1.5$ Hz, 4H), 7.44 (td, $J = 7.5, 1.7$ Hz, 4H), 7.38-7.31 (m, 2H). ^{13}C NMR (100 MHz, CDCl_3) δ 141.3, 128.8, 127.3, 127.2.

Purified by column chromatography (PE); white solid; 97% yield; ^1H NMR (400 MHz, CDCl_3) δ 7.61-7.55 (m, 2H), 7.49 (d, $J = 8.1$ Hz, 2H), 7.42 (t, $J = 7.6$ Hz, 2H), 7.32 (t, $J = 7.4$ Hz, 1H), 7.27-7.24 (m, 2H), 2.39 (s, 3H). ^{13}C NMR (100 MHz, CDCl_3) δ 141.2, 138.4, 137.0, 129.5, 128.7, 127.0, 126.9.

Purified by column chromatography (10:1 PE/EA); colorless oil; 94% yield; ^1H NMR (400 MHz, CDCl_3) δ 8.85 (s, 1H), 8.64-8.53 (m, 1H), 7.91-7.80 (m, 1H), 7.57 (dd, $J = 5.5, 2.0$ Hz, 2H), 7.47 (ddd, $J = 7.9, 5.5, 2.2$ Hz, 2H), 7.43-7.31 (m, 2H). ^{13}C NMR (100 MHz, CDCl_3) δ 148.5, 148.4, 137.8, 136.7, 134.4, 129.1, 128.1, 127.2, 123.6.

Purified by column chromatography (5:1 PE/EA); light yellow solid; 98% yield; ^1H NMR (400 MHz, CDCl_3) δ 8.02 (s, 1H), 7.85 (s, 1H), 7.65 (d, $J = 7.4$ Hz, 2H), 7.48-7.36 (m, 4H), 7.30 (t, $J = 7.4$ Hz, 1H), 7.16 (t, $J = 2.7$ Hz, 1H), 6.58 (s, 1H). ^{13}C NMR (100 MHz, CDCl_3) δ 142.6, 135.4, 133.5, 128.7, 128.4, 127.5, 126.4, 124.9, 122.0, 119.3, 111.3, 103.1.

Purified by column chromatography (PE); white solid; 92% yield; ^1H NMR (400 MHz, CDCl_3) δ 7.26 (dd, $J = 10.6, 4.4$ Hz, 4H), 7.16 (dd, $J = 5.2, 1.8$ Hz, 6H), 3.95 (s, 2H). ^{13}C NMR (100 MHz, CDCl_3) δ 141.2, 129.1, 128.6, 126.2, 42.1.

Supplementary Figure 75. ¹H and ¹³C NMR spectrum for **6**.

Supplementary Figure 76. ¹H and ¹³C NMR spectrum for **7c**.

Supplementary Figure 77. ¹H and ¹³C NMR spectrum for 8c.

Supplementary Figure 78. ^1H and ^{13}C NMR spectrum for biphenyl.

Supplementary Figure 79. ^1H and ^{13}C NMR spectrum for 4-methyl-1,1'-biphenyl.

Supplementary Figure 80. ¹H and ¹³C NMR spectrum for 3-phenylpyridine.

Supplementary Figure 81. ¹H and ¹³C NMR spectrum for 5-phenyl-1H-indole.

Supplementary Figure 82. ¹H and ¹³C NMR spectrum for diphenylmethane.

REVIEWERS' COMMENTS

Reviewer #1 (Remarks to the Author):

The authors have made a detailed and highly professional revision of the manuscript. The article was significantly improved and can be suggested for publication. This is a multidisciplinary study with novel and impactful results.

Reviewer #2 (Remarks to the Author):

The manuscript has been revised in light of my comments and I now recommend publication. The first sentence of the last paragraph of the right column of page 4 dose still not read very well. I leave it to the editor to modify this sentence,

Reviewer #3 (Remarks to the Author):

The manuscript is well-revised to be much better. Although the catalytic activity is not high enough (not satisfactory), the authors show the Suzuki-Miyaura coupling of various aryl chlorides, affording the corresponding coupling products in high yields. They show synthetic applications using their catalytic system. Eventually, the manuscript reaches the criteria to be published in Nat.Comm. as it is.

Responses to Reviewers' Comments

Reviewer #1

Comment. *The authors have made a detailed and highly professional revision of the manuscript. The article was significantly improved and can be suggested for publication. This is a multidisciplinary study with novel and impactful results.*

Response. Thank you very much for the positive comment. On behalf of my co-authors, we would like to express our great appreciation for your help.

Reviewer #2

Comment 1. *The manuscript has been revised in light of my comments and I now recommend publication. The first sentence of the last paragraph of the right column of page 4 dose still not read very well. I leave it to the editor to modify this sentence.*

Response. Thanks for your very careful review and the positive comment. On behalf of my co-authors, we would like to express our great appreciation for your help.

Reviewer #3

Comment. *The manuscript is well-revised to be much better. Although the catalytic activity is not high enough (not satisfactory), the authors show the Suzuki-Miyaura coupling of various aryl chlorides, affording the corresponding coupling products in high yields. They show synthetic applications using their catalytic system. Eventually, the manuscript reaches the criteria to be published in Nat.Comm. as it is.*

Response. Thank you very much for the positive comment. On behalf of my co-authors, we would like to express our great appreciation for your help.